# Structure-Preserving Network Compression Via Low-Rank Induced Training Through Linear Layers Composition

**Ismail R. Alkhouri**[*]                                                          *alkhour3@msu.edu; ismailal@umich.edu*
*Department of Computational Mathematics, Science & Engineering*
*Michigan State University*
*Department of Electrical Engineering & Computer Science*
*University of Michigan - Ann Arbor*

**Xitong Zhang**[*]                                                                                  *zhangxit@msu.edu*
*Department of Computational Mathematics, Science & Engineering*
*Michigan State University*

**Rongrong Wang**                                                                              *wangron6@msu.edu*
*Department of Computational Mathematics, Science & Engineering*
*Department of Mathematics*
*Michigan State University*

**Reviewed on OpenReview:** *https://openreview.net/forum?id=1KCrVMJoJ9*

## Abstract

Deep Neural Networks (DNNs) have achieved remarkable success in addressing many previously unsolvable tasks. However, the storage and computational requirements associated with DNNs pose a challenge for deploying these trained models on resource-limited devices. Therefore, a plethora of compression and pruning techniques have been proposed in recent years. Low-rank decomposition techniques are among the approaches most utilized to address this problem. Compared to post-training compression, compression-promoted training is still under-explored. In this paper, we present a theoretically-justified technique termed **Lo**w-**R**ank **I**nduced **Tra**ining (LoRITa), that promotes low-rankness through the composition of linear layers and compresses by using singular value truncation. This is achieved without the need to change the structure at inference time or require constrained and/or additional optimization, other than the standard weight decay regularization. Moreover, LoRITa eliminates the need to (*i*) initialize with pre-trained models, (*ii*) specify rank selection prior to training, and (*iii*) compute SVD in each iteration. Our experimental results (*i*) demonstrate the effectiveness of our approach using MNIST on Fully Connected Networks, CIFAR10 on Vision Transformers, and CIFAR10/100 and ImageNet on Convolutional Neural Networks, and (*ii*) illustrate that we achieve either competitive or state-of-the-art results when compared to leading structured pruning and low-rank training methods in terms of FLOPs and parameters drop. Our code is available at `https://github.com/XitongSystem/LoRITa/tree/main`.

## 1 Introduction

In recent years, the rapid progress of machine learning has sparked substantial interest, particularly in the realm of Deep Neural Networks (DNNs), utilizing architectures such as Fully Connected Networks (FCNs) (Rumelhart et al., 1986), Convolutional Neural Networks (CNN) (LeCun et al., 1998), and Transformers (Vaswani et al., 2017). DNNs have demonstrated remarkable performance across diverse tasks, including classification (Han et al., 2022), image reconstruction (Ravishankar et al., 2019), object recognition (Minaee

---

[*]The first two authors contributed equally.

et al., 2021), natural language processing (Vaswani et al., 2017), and image generation (Yang et al., 2023). However, the extensive storage and computational requirements of these models hinder their application in resource-limited devices. Addressing these challenges has become a focal point in recent research efforts within the context of a variety of model compression algorithms (Li et al., 2023; Marinó et al., 2023).

Model compression techniques, based on their core strategies, can be classified into parameter quantization (Krishnamoorthi, 2018), knowledge distillation (Hinton et al., 2015), lightweight model design (Zhang et al., 2018), model pruning (He et al., 2019), and low-rank decomposition (Lin et al., 2018). Post-training low-rank decomposition algorithms for network compression necessitate pre-trained weights (Marinó et al., 2023), whereas existing low-rank training methods require structural alterations to the network architecture (Howard, 2017), the specification of a pre-defined rank during training (Kwon et al., 2023), and/or additional constraints or iterations in the training process (Hawkins et al., 2022). This paper explores an effective technique for low-rank training: Over-parameterizing weight matrices to achieve a low-rank model directly through training from random initialization.

Over-parameterizing weight matrices has been employed in neural networks for various purposes beyond compression, as discussed in more detail in Appendix C. In this paper, we revisit the concept of over-parameterization, providing a detailed exploration of it from the perspective of compression. For clarity, we introduce the term **Lo**w-**R**ank **I**nduced **Tra**ining (LoRITa) to describe the specific version of this technique of interest, which employs over-parameterization of the weight matrices by a linear composition of dense matrices. We show both numerically and theoretically how this technique promotes low-rankness of weight matrices. In addition, we extend this technique to convolutional neural networks. Unlike the well-known Low-Rank Adaptation (LoRA) method (Hu et al., 2021), which focuses on low-rank fine-tuning, the LoRITa approach utilizes dense full-rank factors during training. This method, when paired with post-training compression techniques such as Singular Value Thresholding, eliminates the need to determine or know the (sub)optimal ranks of the weights during training. A key finding of this paper is that the layer-composition-based overparametrization should be combined with weight decay—a commonly used regularizer—to effectively promote low-rankness in models. Next, we summarize our contributions.

- We demonstrate that the simple technique of weight matrix over-parameterization through linear layer composition, when combined with weight-decay, effectively encourages low-rankness during training.

- We provide justification for this observation and analyze its theoretical properties.

- We show through extensive experiments that LoRITa applies to a wide range of network architectures and can be used in combination with various post-training compression methods. The experiments include DNN-based image classification tasks across different FCNs, CNNs, and ViTs architectures, using MNIST, CIFAR10, CIFAR100, and ImageNet datasets.

We note that LoRITa is a structure-preserving technique, meaning it does not alter the network architecture, such as reducing the number of filters in CNNs or pruning redundant nodes. To modify the network structure, LoRITa should be combined with other pruning techniques. In this regard, LoRITa functions as a regularization method, akin to weight decay or dropout, which are commonly used in combination with other regularizations.

Additionally, while LoRITa promotes low rankness during training, actual truncation or compression is performed post-training using Singular Value Truncation (SVT) or other more advanced methods.

Since this paper focuses solely on LoRITa, the reported numerical results are based on LoRITa regularization alone during training, in combination with the simplest SVT for compression. Nevertheless, we observe comparable performance to more sophisticated algorithms, highlighting the benefits of the over-parameterization approach. We leave studying the combinations of LoRITa with other pruning methods (or using more advanced post-training compression approaches) to future studies.

## 2 Related Work

In the past decade, numerous methods have been proposed in the realm of DNN compression, employing techniques such as low-rank decomposition or model pruning. Recent survey papers, such as (Li et al., 2023; Marinó et al., 2023; He & Xiao, 2023), offer comprehensive overviews. This section aims to survey recent works closely related to our method. Specifically, we explore papers focused on (*i*) post-training low-rank compression methods, (*ii*) low-rank training approaches, and (*iii*) structured pruning methods.

### 2.1 Post-training Low-Rank Compression Methods

As an important compression strategy, low-rank compression seeks to utilize low-rank decomposition techniques for factorizing the original trained full-rank DNN model into smaller matrices or tensors. This process results in notable storage and computational savings (refer to Section 4 in (Li et al., 2023)).

In the work by (Yu et al., 2017), pruning methods were combined with SVD, requiring feature reconstruction during both the training and testing phases. Another approach, presented in (Lin et al., 2018), treated the convolution kernel as a 3D tensor, considering the fully connected layer as either a 2D matrix or a 3D tensor. Low-rank filters were employed to expedite convolutional operations. For instance, using tensor products, a high-dimensional discrete cosine transform (DCT) and wavelet systems were constructed from 1D DCT transforms and 1D wavelets, respectively.

The authors in (Liebenwein et al., 2021) introduced the Automatic Layer-wise Decomposition Selector (ALDS) method. ALDS uses layer-wise error bounds to formulate an optimization problem with the objective of minimizing the maximum compression error across layers.

While our method and the aforementioned approaches utilize low-rank decomposition and maintain the network structure in inference, a notable distinction lies in our approach being a training method that promotes low-rankness through the composition of linear layers. It is important to emphasize that any low-rank-based post-training compression technique can be applied to a DNN trained with LoRITa. This will be demonstrated in our experimental results, particularly with the utilization of three singular-value truncation methods.

### 2.2 Low-Rank Promoting Methods

Here, we review methods designed to promote low-rankness in DNN training. These approaches typically involve employing one or a combination of various techniques, such as introducing structural modifications (most commonly under-parameterization), encoding constraints within the training optimization process, or implementing custom and computationally-intensive regularization techniques such as the use of Bayesian estimator (Hawkins et al., 2022), iterative SVD (Gural et al., 2020), or the implementation of partially low-rank training (Waleffe & Rekatsinas, 2020).

The study presented in (Kwon et al., 2023) introduces an approach leveraging SVD compression for overparameterized DNNs through low-dimensional learning dynamics inspired by a theoretical study on deep linear networks. The authors identify a low-dimensional structure within weight matrices across diverse architectures. Notably, the post-training compression exclusively targets the linear layers appended to the FCN, necessitating a specialized initialization. More importantly, a pre-specified rank is required, posing a challenge as finding the optimal combination of ranks for all layers is a difficult problem. Since different layers in the model may have different importance to the performance and should be compressed differently, it requires the rank to be layer-specific. For example, the necessity of using layer-specific rank in LoRA is discussed in (Zhang et al., 2024). However, finding the ranks of all layers can be computationally expensive. In LoRITa, this requirement is not needed, as we theoretically show that standard weight decay encourages low rankness without specifying the rank during training. In comparison, our work shares similarities as it employs a composition of multiple matrices. However, our approach encompasses all weight matrices, attention layer weights, and convolutional layers, providing a more comprehensive treatment of DNN architectures.

The study conducted by (Tai et al., 2015) introduced an algorithm aimed at computing the low-rank tensor decomposition to eliminate redundancy in convolution kernels. Additionally, the authors proposed a method

for training low-rank-constrained CNNs from scratch. This involved parameterizing each convolutional layer as a composition of two convolutional layers, resulting in a CNN with more layers than the original. The training algorithm for low-rank constrained CNNs required enforcing orthogonality-based regularization and additional updates. Notably, their approach did not extend to attention layers in ViTs and fully connected weight matrices. Moreover, during testing, unlike our method, their approach necessitates the presence of additional CNN layers.

In a recent contribution by (Sui et al., 2024), a low-rank training approach was proposed to achieve high accuracy, high compactness, and low-rank CNN models. The method introduces a structural change in the convolutional layer, employing under-parameterization. Besides altering the structure, this method requires low-rank initialization and imposes orthogonality constraints during training.

The authors in (Idelbayev & Carreira-Perpiñán, 2020) proposed the low-rank compression of neural networks (LCNN) method, which uses ADMM to optimize the weight matrices constrained with pre-selected ranks. Training using this method is computationally expensive, as in each iteration, the SVD of each matrix is computed. It is noteworthy that LoRITa, in contrast, not only preserves the existing structure at inference time but also eliminates the need for optimization with complex constraints, relying on the standard training with weight decay.

## 2.3 Structured Pruning Methods

When contrasted with weight quantization and unstructured pruning approaches, structured pruning techniques emerge as more practical (He & Xiao, 2023). This preference stems from the dual benefits of structured pruning: not only does it reduce storage requirements, but it also lowers computational demands. As delineated in a recent survey (He & Xiao, 2023), there exists a plethora of structured pruning methods, particularly tailored for CNNs. As our proposed approach offers both storage and computational reductions, the following methods will serve as our baselines in the experimental results.

These methods belong to six categories. Firstly, we consider regularization-based methods, such as Scientific Control for reliable DNN Pruning (SCOP) (Tang et al., 2020), Adding Before Pruning (ABP) (Tian et al., 2021), and the only train once (OTO) (Chen et al., 2021) methods which introduce extra parameters for regularization. Secondly, we consider methods based on joint pruning and compression, exemplified by the Hinge technique (Li et al., 2020), and the Efficient Decomposition and Pruning (EDP) method (Ruan et al., 2021). Thirdly, we consider activation-based methods such as Graph Convolutional Neural Pruning (GCNP) (Jiang et al., 2022) (where the graph DNN is utilized to promote further pruning on the CNN of interest), Channel Independence-based Pruning (CHIP) (Sui et al., 2021), Filter Pruning via Deep Learning with Feature Discrimination in Deep Neural Networks with Receptive Field Criterion (RFC) (DLRFC) (He et al., 2022)), which utilize a feature discrimination-based filter importance criterion, and Provable Filter Pruning (PFP) (Liebenwein et al., 2019). Next, we consider weight-dependent methods, such as the adaptive Exemplar filters method (EPruner) (Lin et al., 2021), Cross-layer Ranking & K-reciprocal Nearest Filters method (CLR-RNF) (Lin et al., 2022), and Filter pruning using high-rank feature map (HRank) (Lin et al., 2020). Next, we consider a method proposed in (Liu et al., 2022) based on automatically searching for the optimal kernel shape (SOKS) and conducting stripe-wise pruning. Lastly, we consider Reinforcement Learning based methods such as Deep compression with reinforcement learning (DECORE) (Alwani et al., 2022).

## 3 Preliminaries

**Notation:** Given a matrix $\mathbf{A} \in \mathbb{R}^{m \times n}$, SVD factorizes it into three matrices: $\mathbf{A} = \mathbf{U}\mathbf{\Sigma}\mathbf{V}^{\top}$, where $\mathbf{U} \in \mathbb{R}^{m \times m}$ and $\mathbf{V} \in \mathbb{R}^{n \times n}$ are orthogonal matrices, and $\mathbf{\Sigma} \in \mathbb{R}^{m \times n}$ is a diagonal matrix with non-negative real numbers known as singular values, $s_i$. For a matrix $\mathbf{A}$, its Schatten $p$-norm, $\|\mathbf{A}\|_p$, is the $\ell_p$ norm on the singular values, i.e., $\|\mathbf{A}\|_p := (\sum_i s_i^p)^{1/p}$. The nuclear (resp. Frobenius) norm, denoted as $\|\mathbf{A}\|_*$ (resp. $\|\mathbf{A}\|_F$), corresponds to the Schatten $p$-norm with $p = 1$ (resp. $p = 2$). For any positive integer $N$, $[N] := \{1, \ldots, N\}$. We use $\sigma(\cdot)$ and $\text{SoftMax}(\cdot)$ to denote the ReLU and softmax activation functions, respectively.

### 3.1 Fully Connected, Convolutional, & Attention Layers

Consider a DNN-based model with $L$ fully-connected layers for which input $\mathbf{x}$ and output $\mathbf{y}$ are related as:

$$\mathbf{y}(\Theta, \mathbf{x}) = \mathbf{W}_L \ldots \sigma(\mathbf{W}_2 \sigma(\mathbf{W}_1 \mathbf{x})), \tag{1}$$

where $\Theta = \{\mathbf{W}_i, \forall i \in [L]\}$ is the set of all parameters. For the sake of simplicity in notation, we will exclude the consideration of the bias, as it should not affect our conclusion.

For CNNs, the input is a third-order tensor for a multi-channel image, with dimensions $H \times W \times D$, where $H$ is the height, $W$ is the width, and $D$ is the depth (the number of channels). The convolutional kernel is a fourth-order tensor denoted by $\mathbf{K}$, with dimensions $F_H \times F_W \times F_D \times M$. The output of the convolution operation is a third-order tensor $\mathbf{O}$ given as

$$\mathbf{O}(x, y, m) = \sum_{i=0}^{F_H-1} \sum_{j=0}^{F_W-1} \sum_{k=0}^{F_D-1} \mathbf{I}(x+i, y+j, k) \mathbf{K}(i, j, k, m). \tag{2}$$

Here, $x$ and $y$ are the pixel indices and $m$ is the channel index in image $\mathbf{O}$. The total number of output channels is equal to the number of filters $M$. The summation iterates over the spatial dimensions of the filter $(i, j)$ and the input depth $k$.

ViTs consist of multi-head attention and fully connected layers. For one-head attention layer (Vaswani et al., 2017), we have $\mathbf{Y} = \text{SoftMax}\left(\frac{\mathbf{X}\mathbf{W}_Q(\mathbf{X}\mathbf{W}_K)^\top}{\sqrt{d}}\right)\mathbf{X}\mathbf{W}_V$, where $\mathbf{X}$ and $\mathbf{Y}$ are the input and output matrices, respectively. We consider the trainable weights $\mathbf{W}_Q$, $\mathbf{W}_K$, and $\mathbf{W}_V$. Here, $d$ corresponds to the queries and keys dimension (Vaswani et al., 2017).

The goal is to compress the trained weights to minimize storage and computational requirements without compromising test accuracy significantly.

### 3.2 Singular Value Thresholding

For a given matrix $\mathbf{W}$ and its SVD $\mathbf{W} = \mathbf{U}\boldsymbol{\Sigma}\mathbf{V}^\top \in \mathbb{R}^{m \times n}$, its best rank-$r$ approximation can be represented as $\mathbf{W} \approx \mathbf{W}_r := \mathbf{U}_r\boldsymbol{\Sigma}_r\mathbf{V}_r^\top$, where $\mathbf{U}_r \in \mathbb{R}^{m \times r}$ contains the first $r$ columns of $\mathbf{U}$, $\boldsymbol{\Sigma}_r \in \mathbb{R}^{r \times r}$ is a diagonal matrix that include the largest $r$ singular values, and $\mathbf{V}_r^\top \in \mathbb{R}^{r \times n}$ contains the first $r$ rows of $\mathbf{V}^\top$. These are called the $r$-term truncation of $\mathbf{U}$, $\boldsymbol{\Sigma}$, and $\mathbf{V}$.

## 4 Compression with LoRITa

In this section, we start by presenting three simple ways to apply SVT to neural networks. Then, we present the details of the LoRITa technique, followed by discussing the theoretical foundation of our method.

### 4.1 Singular Value Truncation of Trained Weights

**Local Singular Value Truncation (LSVT) of Trained Weights:** We simply apply SVT to each weight matrix independently with a fixed rank $r$.

**Global Singular Value Truncation (GSVT) of Trained Weights:** For DNNs, not every layer holds the same level of importance to the output. Therefore, using the local SVT might not lead to the best compression method. To tackle this problem, we can apply a global singular value truncation strategy. This involves normalizing the singular values of each matrix, i.e., dividing each weight matrix by its largest singular value, and then sorting the singular values of the normalized matrices globally. Subsequently, we decide which singular values to drop based on this global ranking. This approach offers an automated method for identifying the principal components of the entire network.

**Iterative Singular Value Truncation (ISVT) of Trained Weights:** An alternative strategy to global ranking is performance-preserving truncation. In each iteration, we fix the number of parameters to truncate,

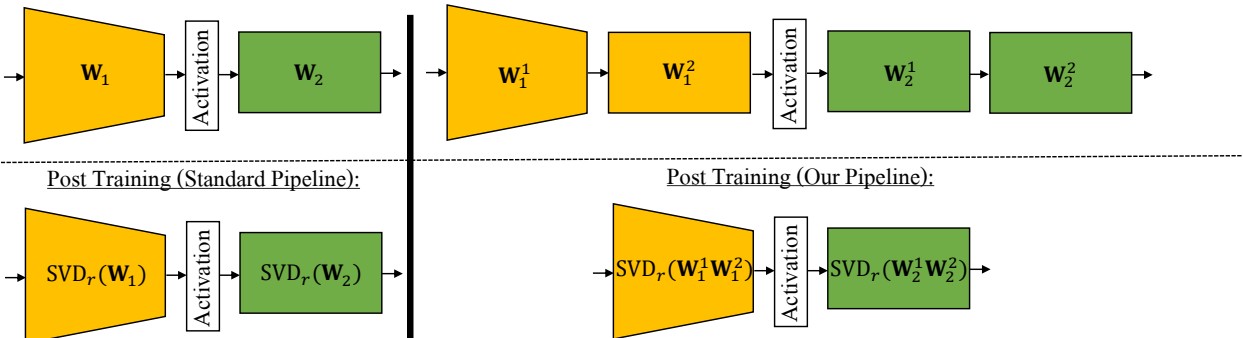

Figure 1: Illustrative example showcasing the standard (*left*) and our (*right*) pipelines. Here, a factorization of 2 is applied to each weight matrix in a simple 2-layered neural network. Post-training, conventional methods employ singular value truncation on the trained weights, whereas in our approach, singular value truncation is conducted on the product of two trained matrices for each weight matrix.

then we decide which layer we will use to remove these parameters by examining the one that induces the lowest increase in the training loss after the truncation. More details are given in Appendix A.

With the $r$-term truncation, for LSVT, GSVT, and ISVT, the storage requirement reduces from $mn$ to $(m+n)r$. We use $\mathrm{SVD}_r(\mathbf{W}) \in \mathbb{R}^{m \times n}$ to denote the SVD $r$-truncated matrix of $\mathbf{W} \in \mathbb{R}^{m \times n}$.

In LSVT, GSVT, and ISVT, our aim is to minimize the accuracy drop resulting from truncation by ensuring that each weight matrix, $\mathbf{W}$, exhibits high compressibility. This compressibility is characterized by a rapid decay in the singular values of $\mathbf{W}$. Such a decay pattern enables us to discard a substantial number of singular values without significantly sacrificing accuracy. Consequently, this property enhances the effectiveness of the compression process. Based on this discussion, a notable insight emerges:

> The faster the singular values of $\mathbf{W}$ decay, the more favorable the compression outcome using SVD. This observation leads to the desire for $\mathbf{W}$ to be of low rank, as low-rank matrices exhibit a faster decay in singular values. The question that naturally arises is: *How to enforce low-rankness in $\mathbf{W}$ during training while preserving the DNN structure at the inference phase?*

Next, we propose our method of low-rank promoted training through the composition of linear layers before activation.

## 4.2 Model Re-parmeterization with LoRITa

We employ the following simple compression pipeline:

- During training, we employ LoRITa to achieve maximal decay of the singular values of the weight matrices in a global manner without sacrificing the model's capacity.

- Post training, we use the LSVT, GSVT or ISVT on the trained weights.

More specifically, we perform the following steps:

- We first express each trainable weight matrix $\mathbf{W} \in \mathbb{R}^{m \times n}$ as a composition of $N > 1$ matrices: $\mathbf{W} = \mathbf{W}^1 \mathbf{W}^2 \dots \mathbf{W}^N$. Here, $\mathbf{W}^1 \in \mathbb{R}^{m \times n}$ and subsequent matrices $\mathbf{W}^2, \mathbf{W}^3, \dots, \mathbf{W}^N \in \mathbb{R}^{n \times n}$. For the model given in Equation (1), the input/output relation becomes:

$$\mathbf{y}(\Theta, \mathbf{x}) = \prod_{k \in [N]} \mathbf{W}_L^k \dots \sigma\Big( \prod_{k \in [N]} \mathbf{W}_2^k \sigma( \prod_{k \in [N]} \mathbf{W}_1^k \mathbf{x}) \Big), \tag{3}$$

where $\Theta = \{\mathbf{W}_i^k, \forall k \in [N], \forall i \in [L]\}$.

---

**Algorithm 1** Compression with LoRITa+SVT.

---

**Input**: $L$ trainable weights $\mathbf{W}_i, \forall i \in [L]$, factorization parameter $N > 1$, and singular value truncation parameter $r$.

 **Output**: Compressed and trained Weights.

1: **For each** $i \in [L]$

2:    **Replace** $\mathbf{W}_i$ by $\mathbf{W}_i^1, \ldots, \mathbf{W}_i^N$

3: **Train** $\mathbf{W}_i^k, \forall i \in [L], \forall k \in [N]$ using Adam and weight decay.

4: **For each** $i \in [L]$

5:    **Use** $\mathrm{SVD}_r(\prod_{k \in [N]} \mathbf{W}_i^k)$ instead of $\mathbf{W}_i$

---

- During training, we minimize the objective

$$\min_{\Theta} \frac{1}{J} \sum_{i \in [J]} \ell(\mathbf{y}(\Theta, \mathbf{x}_i), y_i) + \lambda \sum_{j \in [N]} \sum_{l \in [L]} \|\mathbf{W}_l^j\|_F^2 \, ,$$

  where $\lambda$ is the weight decay parameter, and $J$ is the number of data points (with $(\mathbf{x}_i, y_i)$) in the training set. We note that the proposed factorization works for arbitrary $m$ and $n$.

  Throughout the training process, we optimize the weight matrices $\mathbf{W}_i^k$, for all $i \in [L]$, and $k \in [N]$.

- After the training is finished, we compute the product $\mathbf{W}_i := \prod_{k \in [N]} \mathbf{W}_i^k$, $i \in [L]$, and assign them as the trained weights of the original model.

In summary, LoRITa utilizes the over-parameterized weights during training and reverts to the original weights dimensions after training. The reversion ensures that LoRITa is a structure-preserving technique.

The underlying idea is that a model trained in this manner exhibits enhanced compressibility compared to the original model described by Equation (1). The resulting weight matrices $\mathbf{W}_i$ demonstrate a faster decay in singular values. This strategic approach to training and approximation aims to achieve a more compact and efficient representation of the neural network trained weights, and preserve structure during inference. The procedure is outlined in Algorithm 1. Figure 1 presents an example of our proposed method.

The matrices that correspond to the weights of attention layers are treated in a similar fashion as fully-connected layers.

For convolutional layers, the fourth-order tensor $\mathbf{K}$, with dimensions $F_H \times F_W \times F_D \times M$, is reshaped into a matrix. This is achieved through a mode-4 unfolding, resulting in a matrix $\mathbf{K}^{(4)}$ of size $F_H F_W F_D \times M$. $\mathbf{K}^{(4)}$ is then expressed as a composite of $N$ matrices, denoted as $\mathbf{K}^{(4)} = \mathbf{K}^1 \mathbf{K}^2 \cdots \mathbf{K}^N$. Throughout the training phase, each $\mathbf{K}^i$ is updated as a weight matrix. Post-training, SVD with $r$-truncation is applied to $\mathbf{K}^{(4)} = \mathbf{K}^1 \mathbf{K}^2 \cdots \mathbf{K}^N$ to obtain low-rank factors $\mathbf{L}$ and $\mathbf{R}$, each with $r$ columns, ensuring that $\mathbf{K}^{(4)} \approx \mathbf{L}\mathbf{R}^\top$. During the inference phase, we first compute the convolutions of the input with the filters in $\mathbf{L}$ reshaped back to tensors, then apply $\mathbf{R}^\top$ to the output of the convolutions. More specifically, we reshape $\mathbf{L}$ back to a fourth-order tensor of size $F_H \times F_W \times F_D \times r$, denoted by $\tilde{\mathbf{L}}$, then we conduct its convolution with the input

$$\tilde{\mathbf{O}}(x, y, m') = \sum_{i=0}^{F_H-1} \sum_{j=0}^{F_W-1} \sum_{k=0}^{F_D-1} \mathbf{I}(x+i, y+j, k)\tilde{\mathbf{L}}(i, j, k, m') \, ,$$

with $m' \in [r]$. The final output is obtained as

$$\mathbf{O}(x, y, m) = \sum_{m'=1}^{r} \tilde{\mathbf{O}}(x, y, m')\mathbf{R}(m, m') \, .$$

Compared to the convolution in Equation (2), the computational cost for a single convolution operation is reduced from $\mathcal{O}(F_H F_W F_D M)$ to $\mathcal{O}(F_H F_W F_D r)$. Similarly, the storage requirements are also decreased by a comparable magnitude. Figure 2 illustrates the operations performed with CNNs. Next, We will explain the rationale behind our proposed approach.

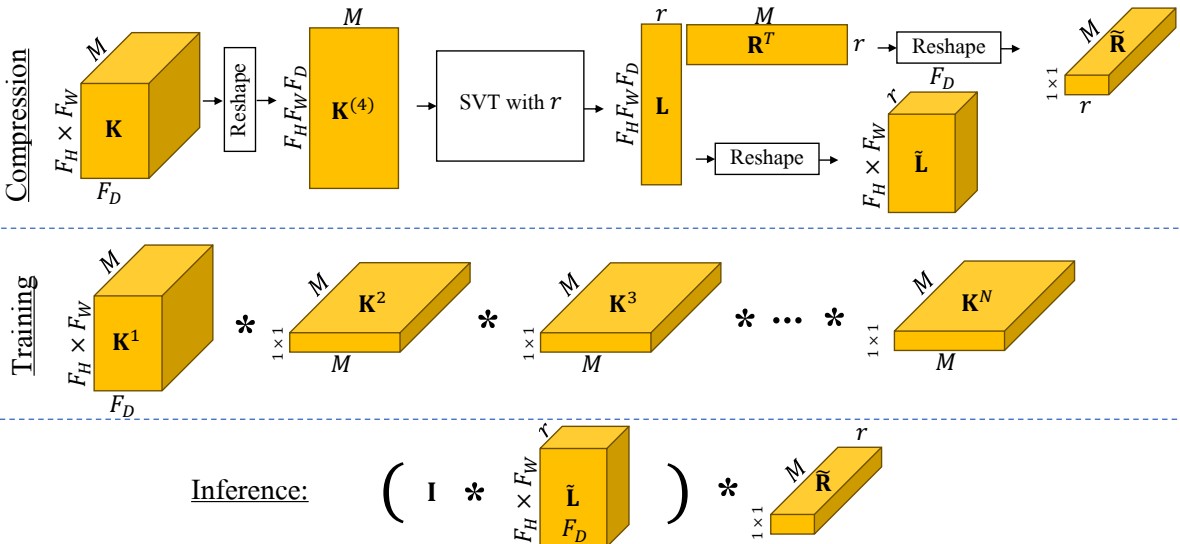

Figure 2: Illustrative block diagram showcasing the compression (*top*), training (*middle*), and inference (*bottom*) for every convolution layer in CNNs. Here, '$*$' denotes the convolution operator.

**Remark 4.1.** *The benefit of over-parameterization in network training has been observed previously in (Khodak et al., 2021; Arora et al., 2018; Guo et al., 2020; Huh et al., 2021) for various reasons. (Arora et al., 2018) proved that over-parameterization can accelerate the convergence of gradient descent in deep linear networks by analyzing the trajectory of gradient flow. (Guo et al., 2020) found that combining over-parameterization with width expansion can enhance the overall performance of compact networks. (Huh et al., 2021) noted through experiments that over-parameterization slightly improves the performance of the trained network, and deeper networks appear to have better rank regularization. These previous works focus on performance and acceleration. Our work complements these findings, providing numerical and theoretical evidence to justify the benefits of over-parameterized weight factorization in network compression due to the weight decay behaving as a rank regularizer. Further discussions for these methods are provided in Appendix C.*

### 4.3 Theoretical Underpinnings of LoRITa

In this subsection, we provide theoretical insight into why LoRITa has to be used with weight-decay.

#### 4.3.1 Weight Decay Regularization & Low-Rankness in LoRITa

Here, we start by stating the following proposition, which says that minimizing the Frobenius norm of the over-parameterized weight matrices is equivalent to minimizing the Schatten p-norm $(0 < p < 1)$ of the original weight matrices and the latter is well-known to promote the low-rankness. The proof of this proposition is given in Appendix B.1.

**Proposition 4.2.** *Let $\mathbf{A} \in \mathbb{R}^{m \times n}$ be an arbitrary matrix and $r \leq \min\{m, n\}$ be its rank. For a fixed integer $N \in \mathbb{Z}_+$, $\mathbf{A}$ can be expressed as the product of $N$ matrices $\mathbf{R}_i \in \mathbb{R}^{m_i \times n_i}$ i.e., $\mathbf{A} = \prod_{i \in [N]} \mathbf{R}_i$ $(m_{i+1} = n_i \geq r,$ $i = 1, ..., N-1$, $m_1 = m$, $n_N = n)$, in infinitely many ways. For any $p \in (0, 1]$ and $p_i > 0$, $\forall i \in [N]$, satisfying $\sum_{i \in [N]} \frac{1}{p_i} = \frac{1}{p}$, it holds*

$$\|\mathbf{A}\|_p = \min_{\mathbf{R}_i, i \in [N]} \left( p \sum_{i \in [N]} \frac{1}{p_i} \|\mathbf{R}_i\|_{p_i}^{p_i} \right)^{1/p} \quad s.t. \quad \prod_{i \in [N]} \mathbf{R}_i = \mathbf{A} . \tag{4}$$

Proposition 4.2 slightly extends Theorems 4 and 5 from (Shang et al., 2016), in its requirement on the dimensions of the factors $\mathbf{R}_i$. Furthermore, we do not assume any prior knowledge of the rank of $\mathbf{A}$, which is conventionally set to its upper bound $\min\{m, n\}$.

Proposition 4.2 indicates that the Schatten-$p$ norm of any matrix is equivalent to the minimization of the weighted sum of Schatten-$p_i$ norm of each factor matrix by which the weights of these terms are $\frac{p}{p_i}$ for all $i \in [N]$.

For $p = 1$, $N = 2$, and $p_1 = p_2 = 2$, Equation (4) reduces to

$$\|\mathbf{A}\|_* = \min_{\mathbf{R}_1, \mathbf{R}_2} \frac{1}{2}\big(\|\mathbf{R}_1\|_F^2 + \|\mathbf{R}_2\|_F^2\big) \quad \text{s.t.} \quad \mathbf{R}_1 \mathbf{R}_2 = \mathbf{A} \,. \tag{5}$$

For some integer $q > 1$, let $p = \frac{1}{q}$, $N = 2q$, and $p_1 = p_2 = \cdots = p_{2q} = 2$. Then, Equation (4) becomes:

$$\|\mathbf{A}\|_{1/q} = \min_{\mathbf{R}_i, i \in [2q]} \Big(\frac{1}{2q} \sum_{i \in [2q]} \|\mathbf{R}_i\|_F^2\Big)^q \quad \text{s.t.} \quad \prod_{i \in [2q]} \mathbf{R}_i = \mathbf{A} \,. \tag{6}$$

If $p \in (0, 1]$, minimizing the Schatten $p$-norm encourages low-rankness. A smaller $p$ strengthens the promotion of sparsity by the Schatten $p$-norm (Nie et al., 2012). Think of the $\mathbf{A}$ matrix in Equation (5) and Equation (6) as weight matrices in FCNs, matricized convolutional layers in CNNs, or matrices representing query, key, and value in attention layers. These identities (Equation (5) and Equation (6)) imply that by re-parameterizing the weight matrix $\mathbf{A}$ as a product of $2q$ other matrices $\mathbf{R}_i$, where $i \in [2q]$, and using $\mathbf{R}_i$ (instead of $\mathbf{A}$) as the variable in gradient descent (or Adam), the weight decay on the new variable corresponds to the right-hand side of Equation (6). Additionally, Equation (6) suggests that the more $\mathbf{R}_i$ we use to represent $\mathbf{A}$, the lower rank we obtain for $\mathbf{A}$. This explains why our proposed model in Equation (3) can achieve a lower rank for the weights compared to the traditional formulation in Equation (1).

**Remark 4.3.** *For any weight matrix of size $m$ by $n$, our proposed training method requires training $nmN$ parameters instead of $mn$ parameters where $N$ is the factorization parameter. Yet in practice, it is more efficient than those SVD-based low-rankness promoting techniques since no explicit SVD is needed. Although Proposition 4.2 suggests that increasing $N$ leads to enhancement in the low-rankness of the appended trained weights, in practice, this leads to prolonged training durations and a more nonlinear landscape of the optimization problem. In experiments, we observe that $N = 3$ is usually sufficient to achieve near-optimal performance, under affordable computational cost. Moreover, prioritizing shorter test times for large models is particularly crucial. This consideration is significant, especially given that testing or inference happens continuously, unlike training, which occurs only once or infrequently. This aspect significantly affects user experience, especially on resource-limited devices where compressed models are often deployed.*

**Remark 4.4.** *In contrast to previous works, Proposition 4.2, the foundation of LoRITa, reveals that we do not assume that the weights are strictly low-rank nor require the knowledge of the weight matrix's (approximate) rank during the training phase. Consequently, our approach encourages low-rankness without compromising the network's capacity.*

### 4.3.2 Sufficiency of a Single Weight Decay Parameter in LoRITa

Proposition 4.2 is used for compressing a single matrix. In the case of nonlinear networks, we have to apply it layer by layer. For each layer, a Schatten-$p$ norm regularization of its weight matrix is introduced to the objective function to penalize the rank, together we have $L$ penalty terms,

$$\min_{\overline{\Theta}} \frac{1}{J} \sum_{i \in [J]} \ell(\mathbf{y}(\overline{\Theta}, \mathbf{x}_i), y_i) + \sum_{l \in [L]} \alpha_l \|\mathbf{W}_l\|_p^p \,,$$

where $\overline{\Theta} = \{\mathbf{W}_l, l \in [L]\}$, and $\alpha_l > 0$ for all $l \in [L]$ are the strengths of the penalties. Let us set $p = \frac{1}{K}$ with some even integer $K$ (the smaller the $p$ is set, the stronger it encourages low-rankness). Then for each weight matrix, we apply Proposition 4.2 to re-parameterize the weight matrix into a depth-$K$ deep matrix factorization. This turns the above optimization into the following equivalent form that avoids the computation of SVD on-the-fly.

$$\min_{\Theta} \frac{1}{J} \sum_{i=1}^J \ell(\mathbf{y}(\Theta, \mathbf{x}_i), y_i) + \sum_{l=1}^L \frac{\alpha_l p}{2} \sum_{i=1}^K \|\mathbf{W}_l^i\|_F^2 \,, \tag{7}$$

| Model | Dataset | $N = 1$ | $N = 2$ | $N = 3$ |
|---|---|---|---|---|
| FCN-6 | MNIST | 0.983 | 0.982 | 0.977 |
| FCN-8 | MNIST | 0.983 | 0.982 | 0.977 |
| FCN-10 | MNIST | 0.983 | 0.981 | 0.977 |
| ViT-L8-H1 | CIFAR10 | 0.717 | 0.729 | 0.727 |
| ViT-L8-H4 | CIFAR10 | 0.710 | 0.718 | 0.701 |
| ViT-L8-H8 | CIFAR10 | 0.705 | 0.719 | 0.714 |

(a) **Without** data augmentation.

| Model | Dataset | $N = 1$ | $N = 2$ |
|---|---|---|---|
| CNN - VGG13 | CIFAR10 | 0.919 | 0.922 |
| CNN - ResNet18 | CIFAR10 | 0.928 | 0.927 |
| CNN - VGG13 | CIFAR100 | 0.678 | 0.686 |
| CNN - ResNet18 | CIFAR100 | 0.708 | 0.714 |
| ViT-L4-H8 | CIFAR10 | 0.861 | 0.852 |
| ViT-L8-H8 | CIFAR10 | 0.865 | 0.857 |
| ViT-L16-H8 | CIFAR10 | 0.856 | 0.867 |

(b) **With** data augmentation.

Table 1: Test Accuracy results of standard ($N = 1$) and LoRITa ($N > 1$) models, Here, $N$ represents the number of composed matrices for each layer. When $N = 1$, it reduces to the original non over-parameterized model. When $N \geq 2$, there is actual over-parameterization and can be called LoRITa (our proposed technique).

where $\Theta = \{\mathbf{W}_l^i, l \in [L], i \in [K]\}$.

However, Equation (7) still has too many tuning parameters $\alpha_l, l \in [L]$. Fortunately, for ReLU networks, we can reduce the number of hyper-parameters to 1 as shown in the following Proposition where the proof is deferred to Appendix B.2.

**Proposition 4.5.** *With ReLU activation, the optimization problem in Equation (7) is equivalent to the following single-hyper-parameter problem*

$$\min_{\Theta} \frac{1}{J} \sum_{i=1}^{J} \ell(\mathbf{y}(\Theta, \mathbf{x}_i), y_i) + \lambda \sum_{l=1}^{L} \sum_{i=1}^{K} \|\mathbf{W}_l^i\|_F^2 \, , \tag{8}$$

*with some proper choice of $\lambda$, in the sense that they share the same minimizers.*

**Remark 4.6.** *Proposition 4.5 supports the practice of using a single weight decay parameter during network training, as is commonly done in the literature.*

## 5 Experimental Results

### 5.1 LoRITa Evaluation on FCNs, CNNs, & ViTs

To rigorously evaluate the effectiveness of the LoRITa method in achieving a rank reduction, we first consider FCNs, CNNs, and ViTs on simple datasets (motivated by the composition of their attention layers, which are essentially built from fully connected layers). The evaluation criterion is the reduction in accuracy w.r.t. the percentage of Retained Singular Values.

The latter is defined as the number of retained singular values divided by the total number of singular values per matrix, averaged across all trained matrices. As a result, the drop in accuracy is computed by subtracting the testing accuracy with SVT from the testing accuracy without applying SVT.

A variety of models, datasets, and over-parameterization scenarios are considered, as outlined in Table 1. For the ViT models, the number following 'L' and 'H' represents the number of layers and the number of heads, respectively. For comparison with baselines, we consider the models with $N = 1$ (models without over-parameterization using LoRITa

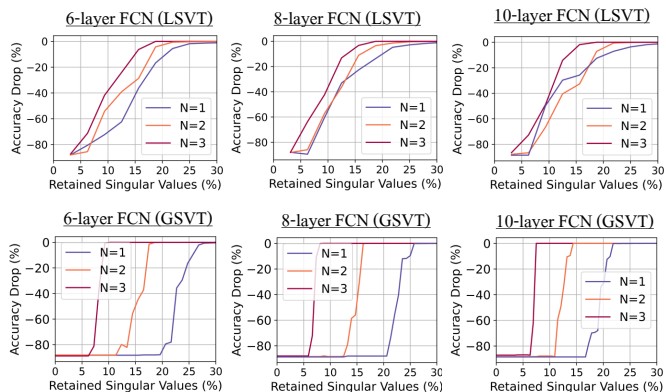

Figure 3: Results of the FCNs in Table 1a. The top (resp. bottom) row corresponds to applying the **Local** (resp. **Global**) SVT. $N = 1$ results represent the baseline, whereas $N > 1$ results are for the LoRITa-trained models.

during training). Further experimental setup details can be found in the caption of Table 1. We use PyTorch to conduct our experiments, and our code will be released at a later date.

In each experiment, our initial phase consists of training the baseline model with optimal weight decay settings to achieve the highest achievable test accuracy. Subsequently, we apply the LoRITa method to re-parameterize the original model. This process involves initializing from a random state and tuning the weight decay to ensure that the final test accuracies remain comparable to those of the initial models. See the test accuracies in Table 1.

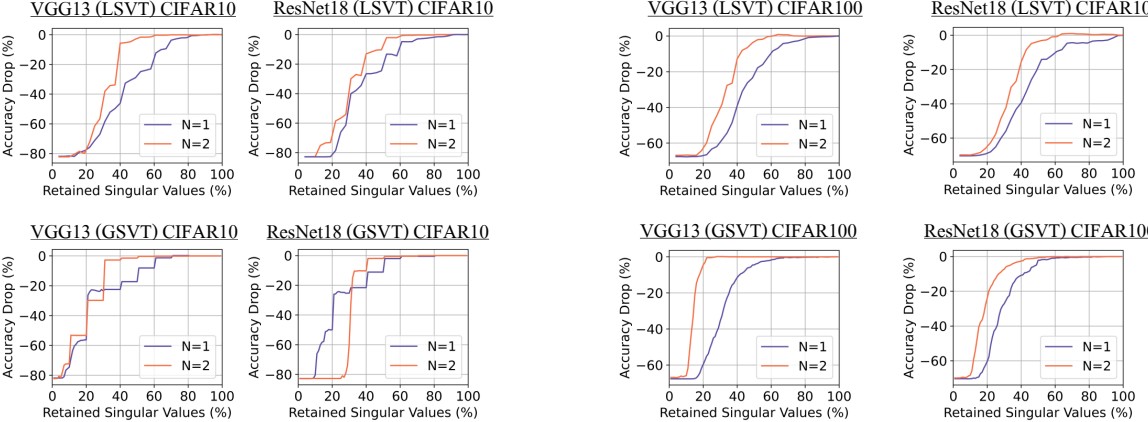

Figure 4: Low-rank compression results of the considered CNNs using the CIFAR10 (*left*) and the CIFAR100 (*right*) datasets for the settings in Table 1b. $N = 1$ denotes the baseline, whereas $N = 2$ represents our method.

Subsequently, we implement SVT on the weight matrices to compress the model, examining the impact on test accuracy as smaller singular values are eliminated. We delve into the LSVT and GSVT strategies, as discussed in Section 4, to determine which singular values to eliminate.

When employing GSVT on ViTs, singular values within the attention and fully connected layers are independently managed.

Similarly, for CNNs, compression is applied distinctly to both the convolutional and fully connected layers, treating each type separately.

First, we evaluate our proposed method on fully connected neural networks, varying the number of layers, utilizing the Adam optimizer with a learning rate set to $1 \times 10^{-2}$, and employing a constant layer dimension of 96 (other than the last). Overparameterization is applied across all layers in the model. To ensure a fair comparison, we begin by tuning the baseline model ($N = 1$) across a range of weight decay parameters $\{5 \times 10^{-6}, 1 \times 10^{-5}, 2 \times 10^{-4}, 5 \times 10^{-5}, 1 \times 10^{-4}, 2 \times 10^{-4}\}$. Subsequently, we extend our exploration of weight decay within the same parameter range for models with $N > 1$. As depicted in Table 1a, setting $N$ to values larger than one results in closely matched final test accuracies. The results for FCNs on the MNIST dataset are illustrated in Figure 3. In these plots, LSVT (resp. GSVT) is employed for the top (resp. bot-

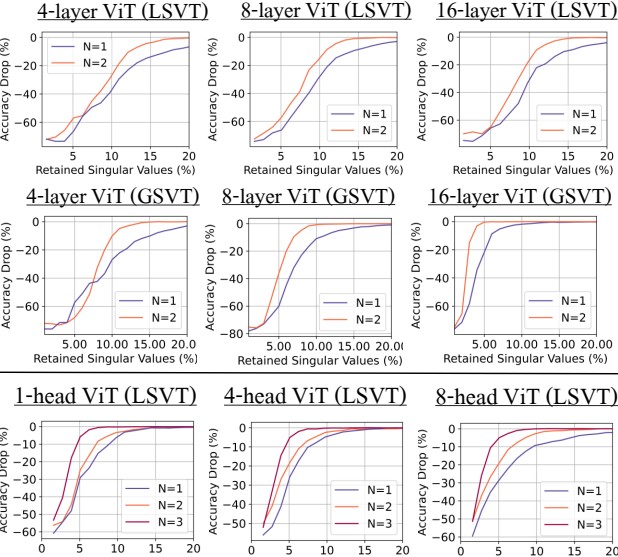

Figure 5: Low-rank compression of ViT models with data augmentation with varied attention heads and layers. The (*top*) and (*bottom*) plots correspond to the settings of Table 1b and Table 1a, respectively. $N = 1$ denotes the baseline, while $N = 2$ and $N = 3$ represent our method.

tom) three plots, providing insights into the effectiveness of the proposed technique. As observed, in almost all the considered cases, models trained with $N > 1$ achieve better compression results (w.r.t. the drop in

test accuracy) when compared to the $N = 1$ models. For the example of the 10-layer FCN model (with GSVT), using LoRITa with $N = 3$ achieves no drop in test accuracy by keeping only 15% of the singular values. When LoRITa is not employed ($N = 1$), the model is required to retain approximately 22% of the singular values in order to achieve the zero drop in test accuracy.

It is important to highlight that the strength of the parameter $\lambda$ associated with the sparsity-promoting regularizer (in this instance, weight decay) is not selected to maximize the compression rate. Instead, the choice of $\lambda$ is geared towards maximizing the test performance of the network, as is conventionally done. Remarkably, adopting this approach still yields a commendable compression rate. In Appendix D, we empirically demonstrate the faster decay of the singular values of the trained weights.

Next, we evaluate the performance of our proposed method on the well-known VGG13 and ResNet18 architectures using the CIFAR10 and CIFAR100 datasets. The learning rate applied in this evaluation is set to $3 \times 10^{-4}$. The weight decay was searched over $\{1 \times 10^{-2}, 5 \times 10^{-3}, 1 \times 10^{-3}\}$ for CIFAR10 and $\{1 \times 10^{-5}, 5 \times 10^{-5}, 1 \times 10^{-4}\}$ for CIFAR100. The results are depicted in Figure 4. Consistent with the FCN results, we observe improved compression outcomes for models trained with LoRITa. The findings from the CNN results suggest that, despite using a straightforward matricization of the convolutional filter (Figure 2), enhanced compression results are still evident. For the instance of VGG13, GSVT, and CIFAR100, retaining 20% of the singular values, not using LoRITa results in 60% drop in accuracy, whereas our LoRITa-trained model only results in nearly 5% drop in test accuracy. Furthermore, we observe that the test accuracy of the compressed model when LoRITa is applied is slightly higher than the test accuracy of the uncompressed model. This is reported at approximately 65% retained singular values when $N = 2$ in Figure 4(*left*).

To validate the efficacy of our compression technique, we evaluate it using ViTs (Beyer et al., 2022), which are noted for their capability to deliver leading-edge image classification outcomes with adequate pre-training. To comprehensively show that our training approach can yield a model of lower rank under various conditions, we first test our method on the CIFAR10 image classification task without data augmentation, employing ViTs with different head counts. Following this, we evaluate our approach on larger ViTs, incorporating data augmentation.

All the considered ViT models underwent optimization via the Adam optimizer with a learning rate of $3 \times 10^{-4}$. The hidden dimension is 256 for all ViTs. To facilitate compression, we apply over-parameterization across all attention modules within the model. For a fair comparison, we initially fine-tuned the baseline model ($N = 1$) across the following weight decay parameters $\{5 \times 10^{-5}, 1 \times 10^{-4}, 2 \times 10^{-4}, 5 \times 10^{-4}, 1 \times 10^{-3}, 2 \times 10^{-3}\}$. Subsequently, we explore weight decay within the same range for our models. As evidenced in Table 1a and Table 1b, setting $N$ to different values results in closely matched final test accuracies.

The compression effects on ViT models without data augmentation are depicted in Figure 5(*top*). We note that with a higher $N$ value, it is possible to achieve a nearly lower model rank. The baseline model's test accuracy begins to decline at an 85% compression rate (retaining 15% of the singular values), whereas the case of $N = 3$ leads to stable test accuracy even at a 95% compression rate (retaining 5% of the singular values). We extend our evaluation to include ViTs with different layers and data augmentation (Figure 5(*bottom*)). The use of data augmentation displayed a considerable improvement in testing accuracy (before compression) as shown when comparing the ViT results of Table 1b and Table 1a. Nevertheless, employing our method consistently resulted in models of a lower rank, as illustrated in Figure 5. The outcomes consistently indicate that our training approach results in models with a lower rank compared to employing only weight decay, across a variety of training configurations and network architectures.

## 5.2 Comparison with Structured Pruning & Low-Rank Training Baselines on CNNs

Here, we compare LoRITa+ISVT with structured pruning and low-rank training techniques delineated in Section 2. The metrics employed for this comparison are the required FLOPs and the number of parameters dropped after compression/pruning, computed using the 'ptflops' Python package[1].

Table 2 and Table 3 present the results for the ResNet20 and VGG-16 architectures, respectively, on the CIFAR10 dataset. Table 4 and Table 5 present the results on ImageNet using ResNet18 and ResNet34

---

[1]https://pypi.org/project/ptflops/

| Method | Acc.(%) (↑) | Pruned Acc.(%) (↑) | Acc. Drop (%) (↓) | Pruned FLOPs (M) (↓) | FLOPs Drop (%) (↓) | Pruned Model Param.(M) (↓) | Param. Drop(%) (↑) |
|---|---|---|---|---|---|---|---|
| LoRITa+ISVT ($N = 3$) | 91.63 | 91 | -0.63 | 18.47 | 54.7 | 0.11 | 61 |
| LoRITa+ISVT ($N = 3$) | 91.63 | 90.54 | -1.09 | 15.44 | 62.3 | 0.084 | 68.8 |
| LoRITa+ISVT ($N = 3$) | 91.63 | 90.17 | -1.46 | 15.06 | 63.2 | 0.078 | **71** |
| LoRITa+ISVT ($N = 2$) | 92.33 | 91.64 | -0.69 | 19.09 | 53.22 | 0.12 | 55.8 |
| LoRITa+ISVT ($N = 2$) | 92.33 | 90.27 | -2.06 | 19.09 | 63.61 | 0.086 | 67.9 |
| Baseline ($N = 1$) | 92.34 | 90.26 | -2.08 | 17.52 | 57.2 | 0.096 | 64.2 |
| Baseline ($N = 1$) | 92.34 | 90.01 | -2.33 | 16.03 | 60.09 | 0.094 | 64.9 |
| SOKS (Liu et al., 2022) | 92.05 | 90.78 | -1.27 | 15.49 | 62.04 | 0.14 | 48.14 |
| SCOP (Tang et al., 2020) | 92.22 | 90.75 | -1.44 | 18.08 | 55.7 | 0.12 | 56.3 |
| Hinge (Li et al., 2020) | 92.54 | 91.84 | -0.7 | 18.57 | 54.5 | 0.12 | 55.45 |
| GCNP Jiang et al. (2022) | 92.25 | 91.58 | -0.67 | 20.18 | 50.54 | 0.17 | 38.51 |
| ABP (Tian et al., 2021) | 92.15 | 91.03 | -1.12 | 21.34 | 47.7 | 0.15 | 45.1 |
| LCNN (Idelbayev & Carreira-Perpinán, 2020) | 91.25 | 90.13 | -0.12 | 13.56 | **66.78** | 0.093 | 65.38 |

Table 2: Evaluation of LoRITa models as compared to SOTA structured pruning/low-rank training methods using **ResNet20** on CIFAR10. The results of the structured pruning methods are reported according to Table 5 in (He & Xiao, 2023) (most recent survey paper) and ranked according to the FLOPs drop percentage. The last row results are reported from Table 1 in (Xiao et al., 2023). The ResNet20 FLOPs (resp. parameters) is 40.81M (resp. 0.27M).

| Method | Acc.(%) (↑) | Pruned Acc.(%) (↑) | Acc. Drop (%) (↓) | Pruned FLOPs (M) (↓) | FLOPs Drop (%) (↓) | Pruned Model Param.(M) (↓) | Param. Drop(%) (↑) |
|---|---|---|---|---|---|---|---|
| LoRITa+ISVT ($N = 3$) | 93.62 | 93.07 | -0.55 | 47.73 | 84.8 | 0.79 | 94.6 |
| LoRITa+ISVT ($N = 3$) | 93.62 | 92.58 | -1.04 | 42.59 | 86.43 | 0.66 | 95.5 |
| LoRITa+ISVT ($N = 3$) | 93.62 | 92.19 | -1.43 | 38.49 | 87.74 | 0.465 | 96.84 |
| LoRITa+ISVT ($N = 3$) | 93.62 | 91.21 | -2.41 | 30.1 | **90.41** | 0.465 | 97.58 |
| LoRITa+ISVT ($N = 2$) | 94.13 | 93.23 | -0.9 | 50.21 | 84.03 | 0.94 | 93.64 |
| LoRITa+ISVT ($N = 2$) | 94.13 | 92.8 | -1.33 | 40.09 | 87.23 | 0.664 | 95.48 |
| LoRITa+ISVT ($N = 2$) | 94.13 | 92.00 | -2.13 | 36.72 | 88.31 | 0.514 | 96.5 |
| Baseline ($N = 1$) | 93.78 | 92.09 | -1.69 | 57.4 | 81.8 | 1.16 | 92.1 |
| Baseline ($N = 1$) | 93.78 | 91.19 | -2.59 | 42.43 | 86.4 | 0.935 | 93.4 |
| DECORE (Setting 1) (Alwani et al., 2022) | 93.96 | 91.68 | -2.28 | 36.95 | 88.25 | 0.26 | **98.26** |
| DECORE (Setting 2) (Alwani et al., 2022) | 93.96 | 92.44 | -1.22 | 51.34 | 83.68 | 0.5 | 96.6 |
| PFP (Liebenwein et al., 2019) | 92.89 | 92.39 | -0.5 | 47.09 | 85.03 | 0.84 | 94.32 |
| OTO (Chen et al., 2021) | 91.6 | 91 | -0.6 | 51.28 | 83.7 | 0.37 | 97.5 |
| ABP (Tian et al., 2021) | 93.96 | 92.65 | -1.31 | 52.81 | 83.21 | 1.5 | 89.79 |
| EDP (Ruan et al., 2021) | 93.6 | 93.52 | -0.08 | 62.57 | 80.11 | 0.65 | 95.59 |
| CHIP (Sui et al., 2021) | 93.96 | 93.18 | -0.78 | 67.32 | 78.6 | 1.87 | 87.3 |
| DLRFC (He et al., 2022) | 93.25 | 93.64 | -0.39 | 72.51 | 76.95 | 0.83 | 94.38 |
| HRank (Lin et al., 2020) | 93.96 | 91.23 | -2.73 | 73.9 | 76.51 | 1.75 | 88.12 |
| EPruner (Lin et al., 2021) | 93.02 | 93.08 | 0.06 | 74.42 | 76.34 | 1.65 | 88.8 |
| CLR-RNF (Lin et al., 2022) | 93.02 | 93.32 | 0.3 | 81.48 | 74.1 | 0.74 | 95 |
| LCNN (Idelbayev & Carreira-Perpinán, 2020) | 92.78 | 92.72 | -0.06 | 45.71 | 85.47 | 1.45 | 91.14 |

Table 3: Evaluation of LoRITa models as compared to SOTA structured pruning/low-rank training methods using **VGG16** on CIFAR10. The results of the structured pruning methods are reported according to Table 3 in (He & Xiao, 2023) (most recent survey paper) and ranked according to the FLOPs drop percentage. The last row results are reported from Table 1 in (Xiao et al., 2023). The VGG16 FLOPs (resp. parameters) is 314.59M (resp. 14.73M). Settings 1 and 2 for DECORE correspond to using different hyper-parameters including the penalty on incorrect predictions (Alwani et al., 2022).

architectures, respectively. The columns denote: 1) Test accuracy before pruning (%); 2) Test accuracy after pruning/compression (%); 3) Accuracy Drop (%); 4) FLOPs after pruning/compression (M); 5) FLOPs drop (%); 6) Pruned model parameters (M); 7) Parameters drop (%).

For CIFAR10 (resp. ImageNet), the results of the structure pruning baselines are as reported in Table 3 and Table 5 (resp. Table 18 and Table 19) of (He & Xiao, 2023), with arrows indicating preferable results. The results of the low-rank training baseline, LCNN Idelbayev & Carreira-Perpinán (2020), are reported from Table 1 in (Xiao et al., 2023).

For the baseline results, we remark that there could be different reasons for not using the same compression ratio: (*i*) Different goals: Reducing the number of parameters and reducing the number of FLOPs are two distinct goals, albeit related. Some papers may focus on one of these objectives, while others aim to achieve a balance between both; (*ii*) Compression is not continuous: In the low-rank case, each time a singular value is eliminated, we remove $m + n$ parameters where $m$ and $n$ are the lengths of the left and right singular vectors of that layer, respectively. Other methods may try to remove filters in CNN, which also has jumps

| Method | Acc.(%) (↑) | Pruned Acc.(%) (↑) | Acc. Drop (%) (↓) | Pruned FLOPs (M) (↓) | FLOPs Drop (%) (↓) | Pruned Model Param.(M) (↓) | Param. Drop(%) (↑) |
|---|---|---|---|---|---|---|---|
| LoRITa+ISVT ($N = 3$) | 67.82 | 66.47 | -1.35 | 1.01 | 44.2 | 6.17 | **47.22** |
| LoRITa+ISVT ($N = 2$) | 69.22 | 68.14 | -1.08 | 1.08 | 40.33 | 6.67 | 42.94 |
| SOKS (Liu et al., 2022) | 70.42 | 69.16 | -1.26 | 0.817 | **54.95** | 6.27 | 46.36 |
| SCOP (Tang et al., 2020) | 69.76 | 69.18 | -0.58 | 1.11 | 38.8 | 7.1 | 39.30 |
| SCOP (Tang et al., 2020) | 69.76 | 68.62 | -1.14 | 0.997 | 45 | 6.6 | 43.5 |
| ABP (Tian et al., 2021) | 70.29 | 67.83 | -2.46 | 1.021 | 43.7 | 6.31 | 46 |

Table 4: Evaluation of LoRITa models as compared to SOTA structured pruning/low-rank training methods using **ResNet18** on ImageNet. The results of the structured pruning methods are reported according to Table 5 in (He & Xiao, 2023) (most recent survey paper) and ranked according to the FLOPs drop percentage. The ResNet18 FLOPs (resp. parameters) is 1.81G (resp. 11.69M).

| Method | Acc.(%) (↑) | Pruned Acc.(%) (↑) | Acc. Drop (%) (↓) | Pruned FLOPs (M) (↓) | FLOPs Drop (%) (↓) | Pruned Model Param.(M) (↓) | Param. Drop(%) (↑) |
|---|---|---|---|---|---|---|---|
| LoRITa+ISVT ($N = 2$) | 72.05 | 71.91 | -0.14 | 1.92 | 47.54 | 11.31 | 48.11 |
| SOKS (Liu et al., 2022) | 74.01 | 73.52 | -0.49 | 1.612 | **55.98** | 11.27 | 48.3 |
| SCOP (Tang et al., 2020) | 73.31 | 72.62 | -0.69 | 2.021 | 44.8 | 11.86 | 45.6 |
| SCOP (Tang et al., 2020) | 73.31 | 72.93 | -0.38 | 2.231 | 39.1 | 13.15 | 39.7 |
| ABP (Tian et al., 2021) | 73.86 | 72.15 | -1.71 | 1.923 | 47.5 | 10.75 | **50.7** |

Table 5: Evaluation of LoRITa models as compared to SOTA structured pruning/low-rank training methods using **ResNet34** on ImageNet. The results of the structured pruning methods are reported according to Table 3 in (He & Xiao, 2023) (most recent survey paper) and ranked according to the FLOPs drop percentage. The ResNet34 FLOPs (resp. parameters) is 3.66G (resp. 21.8M) .

in the ratios; (*iii*) Some compression methods may suddenly break when the compression ratio exceeds a certain threshold, and this threshold varies.

In the tables, $N > 1$ refers to the use of over-parameterization, i.e. LoRITa, and $N = 1$ refers to the classical training of the non-overparameterized network, i.e., the baseline. For both LoRITa+ISVT and the baseline, we apply GSVT, followed by implementing the ISVT approach for which only 120 training data points were used to compute the loss. Apart from using $N = 2$ and $N = 3$, the different results of LoRITa+ISVT correspond to applying ISVT with different desired compression rates (see Appendix A for more details). Here, we use SGD with learning rate $10^{-2}$. Furthermore, it's noteworthy that following all the considered baselines, the reported results for LoRITa+ISVT in Tables 2, 3, 4, and 5 are post one round of fine-tuning, whereas most of the considered baselines employ multiple rounds of fine-tuning of the compressed model. As long as the top-1 accuracy isn't compromised by much (approximately 2%, as per (He & Xiao, 2023)), larger FLOPs drop signifies better compression methods. It's also imperative to note that the last column, Parameters drop, is vital as it signifies the amount of memory reduction.

For VGG16, our results boast the best FLOPs drop and demonstrate competitive parameter drop. In the case of ResNet20, our results achieve the best parameter drop and FLOPs drop compared to structured pruning methods. When compared to the low-rank training baseline (LCNN), our method outperforms LCNN in VGG16 while slightly underperforming in ResNet20 in terms of FLOPs drop. However, it is important to mention that LCNN is significantly more expensive to run, as it requires computing the SVD for every matrix at each iteration during training. In addition to the expensive training, the strong penalty term in LCNN prevents it from reaching the same test accuracy as other methods before compression, resulting in non-competitive test accuracy after pruning even with a small compression rate. Moreover, we observe that for both the considered architectures, **with any fixed level of accuracy drop**, our $N = 3$ model, on average, outperforms our $N = 2$ model in terms of both FLOPs drop and parameter drop. Furthermore, LoRITa models achieve improved performance when compared to applying ISVT on the baseline model ($N = 1$).

LoRITa combined with ISVT compression achieves either the best or nearly the best results across the two CNN CFIAR10 architectures. Most other methods achieve competitive results on one architecture. To highlight this point, we include comparison results from the baselines that considered both architectures in Table 6.

| Method | FLOPs **Drop**(%) ResNet20 (↑) | Param. **Drop**(%) ResNet20 (↑) | FLOPs **Drop**(%) VGG16 (↑) | Param. **Drop**(%) VGG16 (↑) |
|---|---|---|---|---|
| LoRITa+ISVT ($N = 3$) | 63.2 | **71** | **90.41** | **97.58** |
| SOKS | 62.04 | 48.14 | 72.2 | 78.33 |
| Hinge | 54.5 | 55.45 | 39.07 | 80.5 |
| GCNP | 50.54 | 38.51 | 73.07 | 93.06 |
| ABP | 47.7 | 45.1 | 83.21 | 89.71 |
| PFP | 45.46 | 62.67 | 85.03 | 94.32 |
| LCNN | 66.78 | 65.38 | 85.47 | 91.14 |

Table 6: Evaluation of LoRITa as compared to SOTA structured pruning/low-rank training methods that considered ResNet20 and VGG16. The results of the structured pruning methods are reported according to Table 3 in (He & Xiao, 2023) (most recent survey paper) and ranked according to the FLOPs drop percentage. The last row results are reported from Table 1 in (Xiao et al., 2023).

For ImageNet results on ResNet18, we achieve the best parameters drop while reporting slightly lower FLOPs drop when compared to the second best results. For ResNet34, LoRITa+ISVT reports the second best FLOPs drop and lightly lower parameters drop when compared to the second best results with only -0.14% of pruned test accuracy drop.

As mentioned in the introduction section, our numerical experiments focus on examining the effect of LoRITa, so the reported results are based on LoRITa regularization alone with the simplest SVT post-training compression. Across different architectures and datasets, our performance is competitive with other more sophisticated methods. It is possible to further boost the performance when LoRITa is used in combination with other pruning methods and/or with more advanced post-training compression methods.

# 6 Conclusion & Future Work

In this study, we studied a compression technique, **Lo**w-**R**ank **I**nduced **Tra**ining (LoRITa). This theoretically-justified technique promotes low-rankness through the composition of linear layers and achieves compression by employing simple singular value truncation. Notably, LoRITa accomplishes this without necessitating changes to the model structure at inference time, and it avoids the need for constrained or additional optimization steps. Furthermore, LoRITa eliminates the requirement to initialize with full-rank pre-trained models or specify rank selection before training. Our experimental validation, conducted on a diverse range of architectures and datasets, attests to the effectiveness of the proposed approach. Through rigorous testing, we have demonstrated that LoRITa combined with an iterative singular value truncation yields compelling results in terms of model compression and resource efficiency, offering a promising avenue for addressing the challenges associated with deploying deep neural networks on resource-constrained platforms.

In future works, we plan to explore more efficient fine-tuning methods for LoRITa-compressed models for which larger models such as LLMs are considered.

## Acknowledgements

The work was supported by NSF CCF-2212065 and NSF BCS-2215155. The authors would like to thank Avrajit Ghosh (Michigan State University) for insightful discussions.

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

# Appendix

In this Appendix, we first provide a detailed description of the ISVT approach used (Appendix A). Then, we present the proofs in Appendix B. In Appendix C, we review recent works that use linear composition for non-compression methods, followed by demonstrating the faster decay of singular values in LoRITa-trained networks (Appendix D).

## A    Detailed Description of the Iterative SVT Approach

Here, we provide the exact implementation of the ISVT approach we employ in our paper. In each iteration, we fix the number of parameters to truncate. Then, we decide which layer for which these parameters are to be removed by examining which induces the lowest increase in the training loss after the truncation. In particular, for each iteration, we begin by fixing the number of parameters to truncate, setting this number to 500. Next, we select a layer, denoted as $l$, and remove the smallest singular values from this layer until we have moved 500 parameters. After this truncation, we compute the training loss and store this loss as $E(l)$. This process is repeated for all layers to identify the layer $l$ that results in the smallest training loss $E(l)$. Once this layer is identified, we truncate 500 parameters from it. We then proceed to the next iteration, truncating another 500 parameters. This iterative process continues until the desired compression rate is reached. To alleviate the computational cost, we use only 120 randomly subsampled training data to compute $E(l)$.

## B    Proofs

### B.1    Proof of Proposition 4.2

*Proof of Proposition 4.2.* By Theorem 5 of (Shang et al., 2016), we have

$$
\|\mathbf{A}\|_p = \min_{\mathbf{R}_i, i \in [N]} \left( p \sum_{i \in [N]} \|\mathbf{R}_i\|_{p_i}^{p_i} / p_i \right)^{1/p}
$$
$$
\text{s.t.} \prod_{i \in [N]} \mathbf{R}_i = \mathbf{A} ,
$$

$(9)$

provided that

$$\{\mathbf{R}_i\}_{i=1}^N \in \mathcal{T} := \{\{\mathbf{T}_i\}_{i=1}^N : \mathbf{T}_i \in \mathbb{R}^{r \times r}, i = 2, ..., N-1, \mathbf{T}_1 \in \mathbb{R}^{n \times r}, \mathbf{T}_N \in \mathbb{R}^{r \times m}\},$$

with $r$ being the rank of $\mathbf{A} \in \mathbb{R}^{m \times n}$. We want to show that this result still holds if we replace the set $\mathcal{T}$ with

$$\mathcal{T}' := \{\{\mathbf{T}_i\}_{i=1}^N : \mathbf{T}_i \in \mathbb{R}^{m_i \times n_i}, n_i, m_i \geq r, \text{ and } m_{i+1} = n_i \text{ for } i = 1, ..., N-1, n_N = n, m_1 = m\}.$$

In other words, we want to show that

$$
\min_{\substack{\{\mathbf{R}_i\}_{i=1}^N \in \mathcal{T}, \\ \prod_{i \in [N]} \mathbf{R}_i = \mathbf{A}}} \left( \sum_{i \in [N]} \|\mathbf{R}_i\|_{p_i}^{p_i} / p_i \right)^{1/p} = \min_{\substack{\{\mathbf{R}_i\}_{i=1}^N \in \mathcal{T}', \\ \prod_{i \in [N]} \mathbf{R}_i = \mathbf{A}}} \left( \sum_{i \in [N]} \|\mathbf{R}_i\|_{p_i}^{p_i} / p_i \right)^{1/p} ,
$$

where the only distinction between the optimizations on the left and right hand sides of this equality lies in substituting the set $\mathcal{T}$ under the min operator with $\mathcal{T}'$, so that we allow a more flexible choice of dimensions of the factors. We will prove this equality by separately proving

$$
\min_{\substack{\{\mathbf{R}_i\}_{i=1}^N \in \mathcal{T}, \\ \prod_{i \in [N]} \mathbf{R}_i = \mathbf{A}}} \left( \sum_{i \in [N]} \|\mathbf{R}_i\|_{p_i}^{p_i} / p_i \right)^{1/p} \geq \min_{\substack{\{\mathbf{R}_i\}_{i=1}^N \in \mathcal{T}', \\ \prod_{i \in [N]} \mathbf{R}_i = \mathbf{A}}} \left( \sum_{i \in [N]} \|\mathbf{R}_i\|_{p_i}^{p_i} / p_i \right)^{1/p}
$$

$(10)$

and

$$\min_{\substack{\{\mathbf{R}_i\}_{i=1}^N \in \mathcal{T}, \\ \prod_{i\in[N]} \mathbf{R}_i = \mathbf{A}}} \Big( \sum_{i\in[N]} \|\mathbf{R}_i\|_{p_i}^{p_i}/p_i \Big)^{1/p} \leq \min_{\substack{\{\mathbf{R}_i\}_{i=1}^N \in \mathcal{T}', \\ \prod_{i\in[N]} \mathbf{R}_i = \mathbf{A}}} \Big( \sum_{i\in[N]} \|\mathbf{R}_i\|_{p_i}^{p_i}/p_i \Big)^{1/p}. \tag{11}$$

To prove Equation (10), let $\{\mathbf{R}_i^*\}_{i=1}^T \in \mathcal{T}$ be a global minimizer of the optimization on the left hand side of Equation (10). We denote by $\widetilde{\mathbf{R}}_i^*$ the augmented matrices obtained by padding/appending all-zero rows and columns to each $\mathbf{R}_i^*$ until its dimension grows from $r \times r$ to $m_i \times n_i$. Then, $\{\widetilde{\mathbf{R}}_i^*\}_i^N$ is a point in $\mathcal{T}'$. In addition, since the 0-padding does not change the product nor the Schatten $p$-norm of the matrices, plugging $\{\mathbf{R}_i^*\}_{i=1}^T$ into the left hand side optimization of Equation (10) and $\{\widetilde{\mathbf{R}}_i^*\}_{i=1}^T$ into the right hand side one yields to:

$$\min_{\substack{\{\mathbf{R}_i\}_{i=1}^N \in \mathcal{T}, \\ \prod_{i\in[N]} \mathbf{R}_i = \mathbf{A}}} \Big( \sum_{i\in[N]} \|\mathbf{R}_i\|_{p_i}^{p_i}/p_i \Big)^{1/p} = \Big( \sum_{i\in[N]} \|\mathbf{R}_i^*\|_{p_i}^{p_i}/p_i \Big)^{1/p} = \Big( \sum_{i\in[N]} \|\widetilde{\mathbf{R}}_i^*\|_{p_i}^{p_i}/p_i \Big)^{1/p}$$

$$\geq \min_{\substack{\{\mathbf{R}_i\}_{i=1}^N \in \mathcal{T}', \\ \prod_{i\in[N]} \mathbf{R}_i = \mathbf{A}}} \Big( \sum_{i\in[N]} \|\mathbf{R}_i\|_{p_i}^{p_i}/p_i \Big)^{1/p},$$

which proves this direction.

To prove Equation (11), with a little abuse of notation, suppose now $\{\mathbf{R}_i^*\}_{i=1}^N \in \mathcal{T}'$ represents a global minimizer of the right hand side of Equation (11). Let $\mathbf{U} \in \mathbb{R}^{m \times r}$ be the matrix containing the top $r$ left singular vectors of $\mathbf{A}$ ($\mathbf{A}$ is of rank-$r$ by assumption), then

$$\mathbf{A} = \mathbf{U}\mathbf{U}^\top \mathbf{A} = \mathbf{U}\mathbf{U}^\top \mathbf{R}_1^* \cdots \mathbf{R}_N^*.$$

Next, we want to construct from $\{\mathbf{R}_i^*\}_{i=1}^N \in \mathcal{T}'$ a new sequence of matrices of different dimensions $\{\widetilde{\mathbf{R}}_i^*\}_{i=1}^T \in \mathcal{T}$ which yields equal or a smaller objective value than $\{\mathbf{R}_i^*\}_{i=1}^N$. The construction proceeds like the following, first define $\widetilde{\mathbf{R}}_1^* := \mathbf{U}\mathbf{U}^\top \mathbf{R}_1^* \mathbf{V}_1$, where $\mathbf{V}_1$ is the matrix containing the first $r$ right eigenvectors of $\mathbf{U}\mathbf{U}^\top \mathbf{R}_1^*$. This definition ensures that $\widetilde{\mathbf{R}}_1^* \in \mathbb{R}^{r \times r}$ and $\|\widetilde{\mathbf{R}}_1^*\|_p \leq \|\mathbf{R}_1^*\|_p$. For $i = 2, ..., N-1$, define $\widetilde{\mathbf{R}}_i^* := \mathbf{V}_{i-1}^\top \mathbf{R}_i^* \mathbf{V}_i \in \mathbb{R}^{r \times r}$, where $\mathbf{V}_{i-1}$ was defined in the previous iteration and $\mathbf{V}_i$ is matrix holding the top-$r$ right singular vectors of $\mathbf{V}_{i-1}^\top \mathbf{R}_i^*$ as columns. For $i = N$, define $\widetilde{\mathbf{R}}_N^* := \mathbf{V}_{N-1}^\top \mathbf{R}_N^* \in \mathbb{R}^{r \times n}$. With this definition, we can verify the following properties for $\widetilde{\mathbf{R}}_i^*$

- $\{\widetilde{\mathbf{R}}_i^*\}_{i=1}^N \in \mathcal{T}$

- $\|\widetilde{\mathbf{R}}_i^*\|_p \leq \|\mathbf{R}_i^*\|_p$, for any $i = 1, ..., N$, since $\mathbf{V}_i$ have orthonormal columns

- $\prod_{i\in[N]} \widetilde{\mathbf{R}}_i^* = \mathbf{A}$, which is due to

$$\begin{aligned} \mathbf{A} &= \mathbf{U}\mathbf{U}^\top \mathbf{A} \\ &= \mathbf{U}\mathbf{U}^\top \mathbf{R}_1^* \cdots \mathbf{R}_N^* \\ &= (\mathbf{U}\mathbf{U}^\top \mathbf{R}_1^* \mathbf{V}_1)(\mathbf{V}_1^\top \mathbf{R}_2^* \mathbf{V}_2)(\mathbf{V}_2^\top \cdots \mathbf{V}_{N-1})(\mathbf{V}_{N-1}^\top \mathbf{R}_N^*) \\ &= \widetilde{\mathbf{R}}_1^* \cdots \widetilde{\mathbf{R}}_N^*. \end{aligned}$$

Plugging $\{\mathbf{R}_i^*\} \in \mathcal{T}'$ and $\{\tilde{\mathbf{R}}_i^*\} \in \mathcal{T}$ into the right and left hand sides of Equation (11) respectively yields,

$$
\min_{\substack{\{\mathbf{R}_i\}_{i=1}^N \in \mathcal{T}, \\ \prod_{i\in[N]}\mathbf{R}_i \leq \mathbf{A}}} \Big( \sum_{i\in[N]} \|\mathbf{R}_i\|_{p_i}^{p_i}/p_i \Big)^{1/p}
$$

$$
\leq \Big( \sum_{i\in[N]} \|\widetilde{\mathbf{R}}_i^*\|_{p_i}^{p_i}/p_i \Big)^{1/p}
$$

$$
\leq \Big( \sum_{i\in[N]} \|\mathbf{R}_i^*\|_{p_i}^{p_i}/p_i \Big)^{1/p}
$$

$$
= \min_{\substack{\{\mathbf{R}_i\}_{i=1}^N \in \mathcal{T}', \\ \prod_{i\in[N]}\mathbf{R}_i = \mathbf{A}}} \Big( \sum_{i\in[N]} \|\mathbf{R}_i\|_{p_i}^{p_i}/p_i \Big)^{1/p} ,
$$

where the first inequality arises because $\widetilde{\mathbf{R}}_i^* \in \mathcal{T}$ may not represent a minimizer, and therefore, its corresponding objective function value would be greater or equal to the minimum. The second inequality is due to the second bullet point above, whereas the final equality is due to the assumption that $\{\mathbf{R}_i^*\} \in \mathcal{T}'$ represents a minimizer of the right hand side of Equation (11). This completes the proof of the second direction. □

### B.2 Proof of Proposition 4.5

*Proof of Proposition 4.5.* The proposition is derived from the scaling ambiguity inherent in the ReLU activation. The the scaling ambiguity allows for the output of the network to remain unchanged when a scalar is multiplied by the weight matrix of one layer and the same scalar is divided by the weight matrix of another layer.

Let $\{\hat{\mathbf{W}}_l^i, i \in [K], l \in [L]\}$, be the minimizer of Equation (7). Then by taking

$$
\lambda = \frac{p}{2} \left( \prod_{i=1}^L \alpha_l \right)^{1/L}, \tag{12}
$$

one can verify that the rescaled weights,

$$
\{\beta_l \hat{\mathbf{W}}_l^i, i \in [K], l \in [L]\} \quad \text{with} \quad \beta_l = \sqrt{\frac{\alpha_l p}{2\lambda}} , \tag{13}
$$

become the minimizer of Equation (8) and the corresponding minimum coincides with the minimum of Equation (7).

Let us first prove that the objective of Equation (8) evaluated at the rescaled weights $\beta_l \hat{\mathbf{W}}_l^i$ coincides with the objective of Equation (7) evaluated at the original weights $\hat{\mathbf{W}}_l^i$. Indeed, since $\prod_{l\in[L]} \beta_l = 1$ (due to the definition of $\beta_l$ and $\lambda$), and the scaling ambiguity, the two networks corresponding to the rescaled weights and the original weights have identical output. Consequently, when we insert the original and rescaled weights to Equation (7) and Equation (8), respectively, the first terms of the objectives are identical. Additionally, with the chosen value of $\beta_l$, direct calculations confirm that the second terms in these objectives are also the same. This indicates that rescaling each weight $\hat{\mathbf{W}}_l^i$ by $\beta_l$ maintains the consistency of the objective values.

Next, we show that $\{\beta_l \hat{\mathbf{W}}_l^i, i \in [K], l \in [L]\}$ is a minimizer (may not be unique) of Equation (8), by contradiction. If it is not one of the minimizers, then there must exist another set of weights $\{\tilde{\mathbf{W}}_l^i, i \in [K], l \in [L]\}$ that achieve lower objective values for Equation (8) . This in turn implies that the reversely rescaled weights $\{\frac{1}{\beta_l}\tilde{\mathbf{W}}_l^i, i \in [K], l \in [L]\}$ by $\beta_l$ must achieve the same low value for the original objective function Equation (7). This contradicts the assumption that $\{\hat{\mathbf{W}}_l^i, i \in [K], l \in [L]\}$ is a minimizer of Equation (7). Thus, the proof is concluded. □

## C   Discussion on Linear Layers Composition Methods For Non-Compression Tasks

Here, we review recent studies that utilize linear composition for non-compression purposes. This means the methods that involve substituting each weight matrix with a sequence of consecutive layers without including any activation functions. The study presented in (Guo et al., 2020) introduced ExpandNet, where the primary objective of the composition is to enhance generalization and training optimization. The authors empirically show that this expansion also mitigates the problem of gradient confusion.

The research conducted in (Khodak et al., 2021) explores spectral initialization and Frobenius decay in DNNs with linear layer composition to enhance training performance. The focus is on the tasks of training low-memory residual networks and knowledge distillation. In contrast to our method, this approach employs under-parameterization, where the factorized matrices are smaller than the original matrix. Additionally, the introduced Frobenius decay regularizes the product of matrices in a factorized layer, rather than the individual terms. This choice adds complexity to the training optimization compared to the standard weight decay used in our approach.

The study conducted in (Huh et al., 2021) provides empirical evidence demonstrating how linear overparameterization of DNN models can improve the generalization performance by inducing a low-rank bias. Unlike our work, they did not consider the role of weight decay in enforcing low-rankness.

## D   Demonstrating the Faster Decay of Singular Values in LoRITa-trained Models

In Figure 6, we empirically demonstrate the faster decay of singular values in LoRITa-trained models. In particular, Figure 6 (*left*) (resp. Figure 6 (*right*)) show the singular values of the first (resp. second) weight matrix of the standard model ($N = 1$) and LoRITa-trained models ($N = 2$ and $N = 3$) for the FCN8 architecture of Table 1a. As observed, models trained with LoRITa exhibit faster decay, and increasing $N$ promotes faster decay.

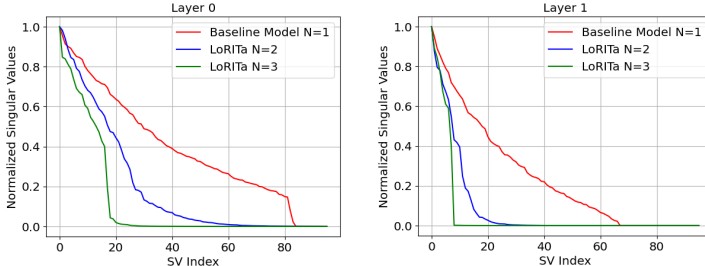

Figure 6: Empirically showing the faster decay of singular values of the first two weight matrices (layer 0 (*left*) and layer 1 (*right*)) of the standard model ($N = 1$) vs. LoRITa-trained models ($N = 2$ and $N = 3$) using the FCN8 architecture of Table 1a.

