# OpenReview forum: "Structure-Preserving Network Compression Via Low-Rank Induced Training Through Linear Layers Composition"
_TMLR — Accepted by TMLR_

### Review · Reviewer_BQ1P · 2024-07-11

**Summary Of Contributions:**

This paper introduced a method to find compressed neural networks via low-rank decomposition of the weight matrices. The proposed method, LoRITa, promotes low-rankness through the composition of linear layers and compresses by using singular value truncation. The authors demonstrate the effectiveness of the proposed approaches compared to many baseline approaches.

**Audience:**

Yes

**Broader Impact Concerns:**

I do not have concerns regarding the broader impacts of this paper.

**Claims And Evidence:**

Yes

**Requested Changes:**

See weaknesses. I would like to see more experimental results to support the proposed approach.

**Strengths And Weaknesses:**

Strengths:
1. The authors provide a lot of technical details on the proposed method LoRITa. I think this has helped me a lot in understanding the paper.
2. The abstract and introduction provides a clear summary of the paper, with a detailed list of contributions.


Weaknesses:
1. I feel the way the authors use to reference LoRITa in the experiments is somewhat confusing. Why using N=2 and N=3 for the proposed approach but N=1 for referencing the dense models.


2. The experimental results are not entirely convincing. From Table 2 and Table 3, the improvement over other baseline approaches seem small. Regarding this part, another question is that why the compression ratio (i.e. parameter drop) is different across methods. This makes it hard to compare between LoRITa and baseline approaches.


3. The authors experimented with ViTs on CIFAR-10, but Vision Transformers are typically used on large-scale datasets as they scale with large amounts of data and compute. I think if the authors want to demonstrate the effectiveness of the proposed approach LoRITa on ViTs, it would be better to evaluate ViTs pre-trained on ImageNet.


4. Rather than using singular value decompositions (SVD) on the weights only, have the authors think of more advanced methods, like reduced rank regression? You can try to optimize a low-rank weight matrix that minimizes the reconstruction error of each linear layer. In principle, this should work better than minimizing the l2 distance of the weights.


5. “Moreover, LoRITa eliminates the need to (i) initialize with pre-trained models, (ii) specify rank selection prior to training”. I don’t understand this part. Could the authors elaborate on this or provide me a reference into a part in the paper that explains this?

---

> ### Author Response · Authors · 2024-09-06
> **Response to Reviewer BQ1P (1 out of 2)**
>
> 1- **I feel the way the authors use to reference LoRITa in the experiments is somewhat confusing. Why using N=2 and N=3 for the proposed approach but N=1 for referencing the dense models.**
>
> Here, $N$ represents the number of composed matrices for each layer.  So when $N=1$, it reduces to the original non over-parameterized  model. Only when $N\geq2$, there is actual over-parameterization and can be called LoRITa  (our proposed method). To address this comment, we included more context to the caption of Table 1 for further clarification.
>
> 2- **The experimental results are not entirely convincing. From Table 2 and Table 3, the improvement over other baseline approaches seem small. Regarding this part, another question is that why the compression ratio (i.e. parameter drop) is different across methods. This makes it hard to compare between LoRITa and baseline approaches.**
>
> We agree that the improvement from the baseline is small. But they still support our main claim that the over-parametrization in LoRITa is useful (when comparing the results with $N>1$ and $N=1$). Please see the global response and the highlighted areas in the revised paper for more details. As mentioned in the global response, we focus on examining the effect of the single LoRITa technique, which enjoys the following merits:
>
> •	Simple and fast implementation: LoRITa training only requires composition of matrices, and the training optimization is standard (with weight decay) and constraints-free.
>
> •	Theoretically grounded (Proposition 4.2): We provide a justification of why the proposed over-parameterization encourages low rankness.
>
> •	One hyper-parameter: Provably one weight decay hyper-parameter across all layers (Proposition 4.5).
>
> •	Applicable to all types of networks: We empirically show the effectiveness of using the proposed approach using MLPs and ViTs, not only CNNs.
>
> All the considered baselines are more sophisticated methods and are generally combinations of different regularizations.
>
> Another important point we’d like to highlight is that LoRITa achieves good results across both CNN architectures. Most other methods achieve competitive results on only one architecture. To highlight this point, we add new results from the baselines in the following table. We include comparison results from baselines that considered both architectures. As observed, our method achieves very competitive flops drop and parameters drop on both architectures.
>
> | Method        | Flops Drop (↑) ResNet20 | Params Drop (↑) ResNet20 | Flops Drop (↑) VGG16 | Params Drop (↑) VGG16 |
> |---------------|---------------------|----------------------|------------------|-------------------|
> | LoRITa (N=3)  | 63.2                | 71                   | 90.41            | 97.58             |
> | SOCKS         | 62.04               | 48.14                | 72.2             | 78.33             |
> | Hinge         | 54.5                | 55.45                | 39.07            | 80.5              |
> | GCNP          | 50.54               | 38.51                | 73.07            | 93.06             |
> | ABP           | 47.7                | 45.1                 | 83.21            | 89.71             |
> | PFP           | 45.46               | 62.67                | 85.03            | 94.32             |
> | LCNN          | 66.78               | 65.38                | 85.47            | 91.14             |
>
> Regarding your question "why the compression ratio (i.e. parameter drop) is different across methods": In Tables 2, 3, 4, and 5 of the revised paper (and those in the original paper), the results of baselines are inherited from the recent survey paper in He & Xiao et al 2023 (most recent survey cited in the captions of the Tables).
>
> We believe that there could be different reasons for not using the same compression ratio. These include:
>
> •	Different goals:  Reducing the number of parameters and reducing the number of FLOPs are two distinct goals, albeit related. Some papers may focus on one of these objectives, while others aim to achieve a balance between both.
>
> •	Compression is not continuous: In the low-rank case, each time we remove a singular value, we remove $(m+n)$ parameters where $m$ and $n$ are the lengths of the left and right singular vectors of that layer. Other methods may try to remove filters in CNN, which also has jumps in the ratios.
>
> •	Some compression methods may suddenly break when the compression ratio exceeds a certain threshold, and this threshold varies. For example, method A may have a 10% accuracy reduction before it breaks while method B only allows a 1% accuracy reduction before it breaks. In this case, one can and tends to report method A with a larger truncation ratio (albeit with a larger drop in test accuracy) than method B.
>
> Therefore, we suggested in the paper to compare rows with similar accuracy drop for their flop drop and/or parameters drop. This discussion is added to Section 5.2 for further clarification.

---

> ### Author Response · Authors · 2024-09-06
> **Response to Reviewer BQ1P (2 out of 2)**
>
> 3- **The authors experimented with ViTs on CIFAR-10, but Vision Transformers are typically used on large-scale datasets as they scale with large amounts of data and compute. I think if the authors want to demonstrate the effectiveness of the proposed approach LoRITa on ViTs, it would be better to evaluate ViTs pre-trained  on ImageNet.**
>
> We agree with the reviewer that ViTs achieve their best results on large-scale datasets. However, the main point of including CIFAR-10 ViT results is to empirically show that our proposed approach is applicable to various deep learning architectures. The ViT models we experimented with achieve test accuracies between 71% and 86.5% (Table 1), which, we feel, is sufficient to empirically show that LoRITa is applicable to models other than CNNs.
>
> Over the past four weeks, we rented two A100 GPUs from Lambda Cloud and attempted to run the requested experiment. However, the process has proven too slow. For VGG19, we are able to complete only 10 epochs per day with a single hyperparameter choice. Despite having just one hyperparameter to tune, the training speed remains insufficient. Training the Vision Transformer (ViT) is even slower. Since our proposed method involves training from scratch, we are unable to use a pre-trained model for ViT.
>
>
>
> 4- **Rather than using singular value decompositions (SVD) on the weights only, have the authors think of more advanced methods, like reduced rank regression? You can try to optimize a low-rank weight matrix that minimizes the reconstruction error of each linear layer. In principle, this should work better than minimizing the l2 distance of the weights.**
>
> We thank the reviewer for their suggestion. We are not exactly sure about the “reduced rank regression”. ***Could the reviewer elaborate more?***
>
> In terms of compressing layer by layer, our paper used a similar approach as we adopted the use of the iterative SVT. In each iteration of ISVT (Section 4 and Appendix A), we start by fixing the number of parameters to truncate, setting this number to 500. We then determine which layer to truncate by evaluating the increase in training loss caused by removing parameters from each layer. Appendix B provides a detailed description of ISVT.
>
> 5-**“Moreover, LoRITa eliminates the need to (i) initialize with pre-trained models, (ii) specify rank selection prior to training”. I don’t understand this part. Could the authors elaborate on this or provide me a reference into a part in the paper that explains this?**
>
> Thank you for your comment. Since different layers in the model may have different importance to the performance and should be compressed differently, it requires the rank to be layer-specific. For example, the necessity of using layer-specific rank in LoRA is mentioned in AutoLoRA: Automatically Tuning Matrix Ranks in Low-Rank Adaptation Based on Meta Learning. However, finding the ranks of all layers can be computationally expensive. In LoRITa, this requirement is not needed, as we theoretically show that standard weight decay encourages low rankness without specifying the rank during training.
>
> Additionally, our results (Figure 3, 4, and 5) showed the need for selecting different rank for different layers, as local SVT compression that uses the same rank across layers is consistently worse than the global SVT compression that uses different rank for every layer.
>
> We added more context to the Introduction and Related Work Sections for further clarifications.
>
> **Requested Changes: See weaknesses. I would like to see more experimental results to support the proposed approach.**
>
> Please see the global response.

---

### Review · Reviewer_JhGQ · 2024-07-21

**Summary Of Contributions:**

This paper proposes a post-training neural network compression approach, i.e., Low-Rank Induced Training (LoRITa). The major aim of LoRITa is efficient inference. LoRITa works by composing linear layers and compressing them using singular value truncation from randomly initialized model weights. Both theoretical justification and experimental results are provided to demonstrate the effectiveness of the proposed LoRITa method.

**Audience:**

Yes

**Broader Impact Concerns:**

I don't find any concerns on the ethical implications of the work.

**Claims And Evidence:**

Yes

**Requested Changes:**

Please see "Weaknesses" above for requested changes. I summarize the items below:
- Compare the proposed LoRITa method with those proposed in [1-3].
- Add MobileNet and EfficientNet experiments.
- Report inference throughput and latency numbers.
- Try LoRITa on larger-scale experiments, e.g., ImageNet and GPT pretraining.

**Strengths And Weaknesses:**

**Strengths:**
- The paper is generally well-written and well-motivated.
- Improving model serving speed and efficiency is a good research direction.
- The proposed LoRITa approach generally makes sense.

**Weaknesses:**
- **[Major]** Similar approaches have been proposed previously, which makes the contribution of this paper a bit limited, but they have not been cited or compared in this paper, e.g., [1-3].
- **[Major]** The conducted experiments are all on a small scale. It is generally understood that convolutional neural networks trained on small image classification tasks, e.g., MNIST and CIFAR, are easy to compress.
- **[Major]** Methods like LoRITa essentially manipulate model architectures. In this sense, how does it compare to MobileNet and/or EfficientNet [4-5]?
- FLOPs do not seem to be a good measurement of inference speed, especially when the data cannot be batchified. What about the latency/throughput of inference?
- Though it is proposed as future work, I strongly believe compressing LLMs is of more interest and practical need compared to compressing CNNs and ViTs.

[1] https://proceedings.mlsys.org/paper_files/paper/2021/file/94cb28874a503f34b3c4a41bddcea2bd-Paper.pdf
[2] https://proceedings.mlsys.org/paper_files/paper/2023/file/c2db3ef0b1ad4c5ec7c3a0a0c6f6c832-Paper-mlsys2023.pdf
[3] https://arxiv.org/abs/2202.00834
[4] https://arxiv.org/abs/1704.04861
[5] https://arxiv.org/abs/1905.11946

---

> ### Author Response · Authors · 2024-09-06
> **Response to Reviewer JhGQ (1 out of 2)**
>
> 1-**[Major] Similar approaches have been proposed previously, which makes the contribution of this paper a bit limited, but they have not been cited or compared in this paper, e.g., [1-3].**
>
> We thank the reviewer for sharing these papers. We would like to point out that there are some major differences between our proposed approach and the suggested references [1] (PUFFERFISH) and [2] (CATTLEFISH), particularly in the training phase. The training in our method does not require (1) estimating the rank, (2) warm-up training on the full rank weights, and (3) computing the singular values during training. In LoRITa, we theoretically and empirically show that standard weight decay is sufficient to encourage low rankness of the composed linear weights during training. The singular value truncation in our method is applied post training.
>
> We would like to share an interesting observation from our experiments that highlights another major difference between LoRITa and previous methods. After training with LoRITa, we determine the optimal rank for each layer by applying Singular Value Thresholding (SVT) to the trained weights, as described in the paper (though other methods can also be used to find these optimal ranks). When these optimal ranks are fed into PUFFERFISH—where the weights are reinitialized with Kaiming initialization, the dimensions of the weight matrices are set to these optimal ranks, and then training is conducted on the U and V factors—the results are noticeably worse compared to training the model with LoRITa, compressing with SVT, and slightly fine-tuning the resulting model.
>
> We believe this difference stems from over-parameterization. LoRITa inherently over-parameterizes the weights, while PUFFERFISH employs under-parameterization by using thin, low-rank factors. In the context of network generalization, it is well established that over-parameterization combined with appropriate regularization can enhance the model's generalization and performance on test datasets. We believe that LoRITa's over-parameterization effectively reduces model sharpness, thus contributing to these performance benefits. This observation appears to be consistent across multiple networks.
>
> As such, we believe our contribution, albeit share some similarities with these methods, still offers meaningful insights and value.
>
> Despite the differences, in the following table, we compare the parameters drop for both methods on different architectures using CIFAR10:
>
> | Method           | Network  | Parameters (M) | Pruned Parameters (M) | Parameters Drop (%) |
> |------------------|----------|----------------|-----------------------|---------------------|
> | LoRITa (N=3)     | ResNet20 | 0.27           | 0.078                 | 71.11               |
> | LoRITa (N=3)     | VGG16    | 14.73          | 0.465                 | 97.68               |
> | PUFFERFISH [1]   | VGG19    | 20.56          | 8.37                  | 59.29               |
> | PUFFERFISH [1]   | ResNet18 | 11.17          | 3.34                  | 70.1                |
> | CATTLEFISH [2]   | VGG19    | 20.56          | 1.9                   | 90.7                |
> | CATTLEFISH [2]   | ResNet18 | 11.17          | 2                     | 82.1                |
>
> Note that we achieve these results without the need to estimate the rank and compute the SVD during training.
>
> For the method proposed in [3], estimating the rank during the training is still needed.
>
> 2- **[Major] The conducted experiments are all on a small scale. It is generally understood that convolutional neural networks trained on small image classification tasks, e.g., MNIST and CIFAR, are easy to compress.**
>
> Please refer to the global response.
>
>
>
> 3-**[Major] Methods like LoRITa essentially manipulate model architectures. In this sense, how does it compare to MobileNet and/or EfficientNet [4-5]?**
>
> While during the training phase, we over-parameterize the network with the proposed linear composition of weights, at the end the training before the SV truncation, we revert the model back to the original architecture. The weights in this original architecture are then assigned as the product of the weights from the composed layers. This means that LoRITa is a ***structure-preserving*** method at inference. See Figure 1 for an illustrative example about how the structure remains unchanged during the testing phase.
>
> On the other hand, the methods in [4] and [5] present a complete re-scaling of the model architectures at training and testing phases. This difference makes these methods not competitors. However, it is important to note that these methods can benefit from the proposed linear composition compbined with weight decay regularization.

---

> ### Author Response · Authors · 2024-09-06
> **Response to Reviewer JhGQ (2 out of 2)**
>
> 4-**FLOPs do not seem to be a good measurement of inference speed, especially when the data cannot be batchified. What about the latency/throughput of inference?**
>
> Thank you for your comment. While we are not hardware experts, our understanding is that FLOPs represent a theoretical limit that any hardware can achieve, making it somewhat hardware-independent, as we hoped. According to the survey paper "Structured Pruning for Deep Convolutional Neural Networks: A Survey," FLOPs is a widely used metric in the fields of compression, pruning, knowledge distillation, and quantization to indicate computational complexity.
>
>
> Our understanding is that latency/throughput of inference is directly related to FLOPs. While they also depend on the hardware and availability of parallel processing units, latency generally increases when the required FLOPs are high and visa versa. We understand that FLOPs do not take hardware implementation into account. However, if an improved hardware implementation is available, reducing FLOPs would also lead to improved results.
>
> 5-**Though it is proposed as future work, I strongly believe compressing LLMs is of more interest and practical need compared to compressing CNNs and ViTs.**
>
> We agree and acknowledge the reviewer’s opinion/suggestion as many recent practical uses of deep learning models are for LLMs. Please note that our method is a training method, which means that we would need to train LLMs from scratch. While we are confident that LoRITa can be adopted to LLMs (as our method is theoretically supported), for the time being, we do not have the compute power to conduct such experiments. As such, for now, we leave experimenting with LLM models for future work.
>
> We believe that the results from our current experiments are adequate to support the revised claim. Please refer to the global response and the introduction in the revised paper for further details.

---

### Review · Reviewer_wD5i · 2024-07-29

**Summary Of Contributions:**

In this manuscript, the authors propose a novel training scheme called LoRITa, which leverages the composition of linear layers and compression via singular value truncation. Using datasets such as MNIST and CIFAR-10/100, the proposed method demonstrates competitive accuracy for Vision Transformers and CNN models. The experimental results obtained with the proposed scheme are compared against established structured pruning and low-rank training methods.

**Audience:**

Yes

**Claims And Evidence:**

Yes

**Requested Changes:**

Please refer to weakness part

**Strengths And Weaknesses:**

[Strengths]
- The method is straightforward, and the manuscript is easy to read.
- The mathematical formulation in Section 4 is clear.
- Despite N being limited to 3, Table 1 and Figures 3, 4, and 5 effectively present the motivation for the proposed work.

[Weaknesses]
- The manuscript does not address the computational overhead for training, particularly when N becomes large. Comparisons may be unfair if the baseline for structured pruning starts from a relatively smaller model, whereas the proposed method might require additional computations during training.
- The manuscript only uses very small datasets. The authors are strongly encouraged to include experiments using the ImageNet dataset. Without this, it is difficult to assess whether the proposed method is effective beyond small datasets, as CIFAR-10/100 and MNIST are too small compared to ImageNet.
- Previous works on structured pruning or low-rank training methods include experimental results using ImageNet, which the authors do not provide.
- For instance, from Tables 2 and 3, it appears that the proposed method is only effective for small datasets or highly redundant models. Conducting experiments with a combination of Vision Transformer and ImageNet would significantly strengthen the claims.

---

> ### Author Response · Authors · 2024-09-06
> **Response to Reviewer wD5i**
>
> 1-**The manuscript does not address the computational overhead for training, particularly when N becomes large. Comparisons may be unfair if the baseline for structured pruning starts from a relatively smaller model, whereas the proposed method might require additional computations during training.**
>
> We acknowledge the reviewer’s comment. For any weight matrix $m$ by $n$, our proposed training method requires training $nmN$ parameters instead of $mn$ parameters, where $N$ is over-parametrization rate normally set to 2 or 3 in our experiments.  We would like to point out that increasing the number of layers is computationally much more efficient than applying SVD during training which is the standard approach in many compression techniques to promote the low-rankness (see the compression survey we cite in the related work section). In the revised manuscript, we include more context to Remark 4.3 for further clarification regarding this point.
>
> As you correctly pointed out, a full compression pipeline includes both training and post-training compression. Some methods alter the training phase, some modify the compression phase, and some change both. LoRITa specifically modifies only the training phase, and we have only tested it in combination with a basic Singular Value Thresholding (SVT) compression method. Therefore, direct comparisons among these methods may not be entirely fair. We have already excluded comparisons with methods where the disparity was evident. Currently, we do not have an ideal solution for this issue, as comparing different methods solely based on final performance seems to be a common practice in this field.
>
> Additionally, as noted in our global response, our numerical experiments primarily aim to demonstrate that using over-parameterization with LoRITa is more effective than not using it, meaning that $N>1$ yields better results than $N=1$, when all other parts in the pipeline is fixed. Most of our experiments focus on this comparison, which we believe is more reliable.
>
>
>
> 2-**The manuscript only uses very small datasets. The authors are strongly encouraged to include experiments using the ImageNet dataset. Without this, it is difficult to assess whether the proposed method is effective beyond small datasets, as CIFAR-10/100 and MNIST are too small compared to ImageNet.**
>
> Thank you for your comment. Please see the global response.
>
> 3-**Previous works on structured pruning or low-rank training methods include experimental results using ImageNet, which the authors do not provide.**
>
> Thank you for your comment. Please see the global response.
>
> 4-**For instance, from Tables 2 and 3, it appears that the proposed method is only effective for small datasets or highly redundant models. Conducting experiments with a combination of Vision Transformer and ImageNet would significantly strengthen the claims.**
>
> Thank you for your comment. Over the past four weeks, we rented two servers, each with an A100 GPU from Lambda Cloud and attempted to run the Resnet18/ResNet34/Vgg19/Alexnet/ViT + Imagenet experiments, but only ResNet18 and ResNet34 were fast enough to deliver a meaningful result on time. Vgg19 and Alexnet run 10 iterations per day and ViT is even slower.  Despite having just one hyperparameter to tune, the training speed remains insufficient. Since our proposed method involves training, we are unable to use a pre-trained model for ViT.

---

### Author Response · Authors · 2024-09-06
**Global Response 2 out of 2**

- ImageNet with ResNet18:

| Method                         | Acc. (%) (↑) | Pruned Acc. (%) (↑) | Acc. Drop (%) (↓) | Pruned FLOPs (G) (↓) | FLOPs Drop (%) (↑) | Pruned Model Param. (M) (↓) | Param. Drop (%) (↑) |
|---------------------------------|--------------|---------------------|-------------------|----------------------|--------------------|-----------------------------|---------------------|
| LoRITa+ISVT (N=3)               | 67.82        | 66.47               | -1.35             | 1.01                 | 44.2               | 6.17                        | **47.22**           |
| LoRITa+ISVT (N=2)               | 69.22        | 68.14               | -1.08             | 1.08                 | 40.33              | 6.67                        | 42.94               |
| SOKS        | 70.42        | 69.16               | -1.26             | 0.817                | **54.95**          | 6.27                        | _46.36_             |
| SCOP        | 69.76        | 69.18               | -0.58             | 1.11                 | 38.8               | 7.1                         | 39.30               |
| SCOP        | 69.76        | 68.62               | -1.14             | 0.997                | _45_               | 6.6                         | 43.5                |
| ABP       | 70.29        | 67.83               | -2.46             | 1.021                | 43.7               | 6.31                        | 46                  |




- ImageNet with ResNet34:


| Method                        | Acc. (%) (↑) | Pruned Acc. (%) (↑) | Acc. Drop (%) (↓) | Pruned FLOPs (G) (↓) | FLOPs Drop (%) (↑) | Pruned Model Param. (M) (↓) | Param. Drop (%) (↑) |
|-------------------------------|--------------|---------------------|-------------------|----------------------|--------------------|-----------------------------|---------------------|
| LoRITa+ISVT (N=2)              | 72.05        | 71.91               | -0.14             | 1.92                 | _47.54_            | 11.31                        | 48.11               |
| SOKS        | 74.01        | 73.52               | -0.49             | 1.612                | **55.98**          | 11.27                        | _48.3_              |
| SCOP       | 73.31        | 72.62               | -0.69             | 2.021                | 44.8               | 11.86                        | 45.6                |
| SCOP       | 73.31        | 72.93               | -0.38             | 2.231                | 39.1               | 13.15                        | 39.7                |
| ABP      | 73.86        | 72.15               | -1.71             | 1.923                | 47.5               | 10.75                        | **50.7**            |

---

### Author Response · Authors · 2024-09-06
**Global Response 1 out of 2**

#### We would like to thank the AC and reviewers for their constructive comments.

Dear reviewers: the previous writing may have misleadingly suggested that LoRITa is an end-to-end compression algorithm. Technically, LoRITa refers to a regularization technique that involves training with over-parameterization through linear layer composition. It is just one step in the compression pipeline, similar to other regularization methods like dropout and weight decay. LoRITa can and should be used in combination with other techniques and post-compression methods.

In this paper, we focus specifically on LoRITa and evaluate its effectiveness. Our goal is to demonstrate that using this technique within the compression pipeline yields better results than not using it. However, the final performance also depends on other aspects of the pipeline, such as the choice of post-training compression methods and additional regularization techniques. Since LoRITa is structure-preserving, it does not alter the size of kernels or hidden layers, and thus must be combined with other pruning methods to achieve optimal compression. We have revised the contribution section of the introduction (highlighted in blue) to clarify this main message. We faced a choice between focusing solely on the properties of LoRITa versus striving for state-of-the-art (SOTA) results by integrating LoRITa with other regularizations and advanced post-training compression methods. We chose to concentrate on the former, studying LoRITa independently to highlight its broad applicability. Our numerical experiments aim to show that using LoRITa (with $N>1$) provides better results than not using it ($N=1$), given other parts of the pipeline fixed. We test this across various networks and post-training compression methods.

We understand that the reviewers are interested in seeing end results that surpass SOTA, which might stem from our previous writing. Due to LoRITa's structure-preserving nature and our choice not to mix it with other regularizations, our results on CIFAR are on par with SOTA but not noticeably superior. Nonetheless, we are also interested in what LoRITa alone can achieve on ImageNet without combining it with other techniques. Our experiments indicate that while LoRITa achieves nearly SOTA-level parameter reduction, it does not achieve the same level of FLOP reduction. This is not unexpected, as low-rank compression methods based on LR factorization reduce the input dimension through multiplication with L but must revert to the original dimension after multiplication with R. Therefore, methods that directly reduce dimensions or the number of kernels typically achieve greater FLOP reductions. We'd like to point out that state-of-the-art methods, such as SOKS, involve multiple regularizations, different stages, and significantly more hyperparameters. Therefore, as reviewer JhGQ pointed out, the comparison in the table below may not be entirely fair.

Overall, our conclusion is that LoRITa serves as a valuable addition during training, enhancing the efficiency of post-training compression methods.

We included ImageNet Results on ResNet18 and ResNet34 architecture and demonstrated that LoRITa when combined with a simple post-training iterative SVT achieves competitive results. Table 4 and Table 5 of the revised manuscript present these results.

---

### Decision · Action_Editor_Q8B2 · 2024-10-09

**Recommendation:** Accept as is

**Comment:**

One of the major comments raised by the reviewers was the lack of experiments on relatively large datasets and neural networks. In response, the authors conducted additional experiments using the ImageNet dataset. All reviewers recommended acceptance after the authors’ response. I concur with the reviewers: the idea of exploiting the low-rank induced training method for model compression is interesting, and the results demonstrate its potential for model compression, though it does not achieve the best results on ImageNet.

**Audience:**

Researchers working on network compression and efficiency would be interested in this work

**Claims And Evidence:**

This paper introduces a training method for network compression called Low-Rank Induced Training (LoRITa), which promotes low-rankness through the composition of linear layers for the weight matrix in each layer during training, followed by compression using singular value truncation after training. The main idea behind LoRITa is the use of induce low-rank regularization via weight decay for the composition of linear layers, encouraging low-rank weight matrices during training. The authors demonstrate the effectiveness of the proposed approach compared to several baseline methods.